# Automating Data Annotation under Strategic Human Agents: Risks and Potential Solutions

**Tian Xie**
Computer Science and Engineering
the Ohio State University
Columbus, OH 43210
xie.1379@osu.edu

**Xueru Zhang**
Computer Science and Engineering
the Ohio State University
Columbus, OH 43210
zhang.12807@osu.edu

## Abstract

As machine learning (ML) models are increasingly used in social domains to make consequential decisions about humans, they often have the power to reshape data distributions. Humans, as strategic agents, continuously adapt their behaviors in response to the learning system. As populations change dynamically, ML systems may need frequent updates to ensure high performance. However, acquiring high-quality *human-annotated* samples can be highly challenging and even infeasible in social domains. A common practice to address this issue is using the model itself to annotate unlabeled data samples. This paper investigates the long-term impacts when ML models are retrained with *model-annotated* samples when they incorporate human strategic responses. We first formalize the interactions between strategic agents and the model and then analyze how they evolve under such dynamic interactions. We find that agents are increasingly likely to receive positive decisions as the model gets retrained, whereas the proportion of agents with positive labels may decrease over time. We thus propose a *refined retraining process* to stabilize the dynamics. Last, we examine how algorithmic fairness can be affected by these retraining processes and find that enforcing common fairness constraints at every round may not benefit the disadvantaged group in the long run. Experiments on (semi-)synthetic and real data validate the theoretical findings.

## 1 Introduction

As machine learning (ML) is increasingly used to automate human-related decisions (e.g., in lending, hiring, college admission), there is a growing concern that these decisions are vulnerable to human strategic behaviors. With the knowledge of decision policy, humans may adapt their behavior strategically in response to ML models, e.g., by changing their features at costs to receive favorable outcomes. A line of research called *Strategic Classification* studies such problems by formulating mathematical models to characterize strategic interactions and developing algorithms robust to strategic behavior [1, 2]. Among existing works, most studies focus on one-time model deployment where an ML model is trained and applied to a fixed population of strategic agents *once*.

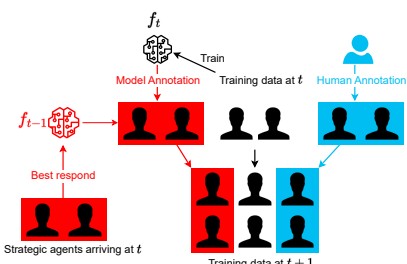

Figure 1: Illustration of updating training data from $t$ to $t+1$ during the retraining process with strategic feedback

However, practical ML systems often need to be retrained periodically to ensure high performance on the current population. As the ML model gets updated, human behaviors also change accordingly.

38th Conference on Neural Information Processing Systems (NeurIPS 2024).

To prevent the potential adverse outcomes, it is critical to understand how the strategic population is affected by the model retraining process. Traditionally, the data used for retraining models can be constructed manually with human annotations (e.g., ImageNet). However, acquiring a large amount of *human-annotated* samples can be highly difficult and even infeasible, especially in human-related applications, e.g., in automated hiring where an ML model is used to identify qualified applicants, even an experienced interviewer needs time to label an applicant.

Motivated by a recent practice of *automating* data annotation for retraining large-scale ML models [3, 4], we study strategic classification in a sequential framework where an ML model is periodically retrained by a decision-maker with both *human* and *model-annotated* samples. These updated models are deployed sequentially on agents who may modify their features to receive favorable outcomes. Since ML models affect agent behavior and the agent strategic feedback can further be captured when retraining the future model, their interactions drive both to change dynamically over time. However, it remains unclear how the two evolve under such dynamics and what long-term effects one may have on the other.

To further illustrate our problem, consider an example of college admission where new students from a population apply each year. In the $t$-th year, an ML model $f_t$ is learned from a training dataset $\mathcal{S}_t$ and used to make admission decisions. For students who apply in the $(t+1)$-th year, they will best respond to the model $f_t$ in the previous year (e.g., preparing the application package in a way that maximizes the chance of getting admitted). Meanwhile, the college

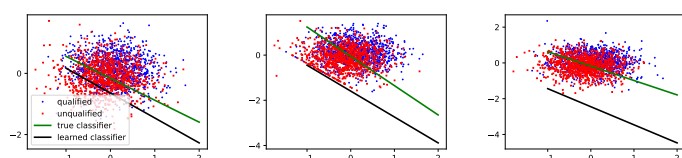

Figure 2: Evolution of the student distribution and ML model at $t = 0$ (left), $t = 5$ (middle), and $t = 14$ (right): each student has two features. At each time, a classifier is retrained with both human and model-annotated samples, and students best respond to be admitted (Fig. 1). Over time, the learned classifier (black lines) deviates from ground truth (green lines).

retrains the classifier $f_{t+1}$ using a new training dataset $\mathcal{S}_{t+1}$ consisting of previous training data $\mathcal{S}_t$, new *human-annotated* samples, and new *model-annotated* samples (i.e., previous applicants annotated by the most recent model $f_t$). The model $f_{t+1}$ is then used to make admission decisions in the $(t+1)$-th year. This process continues over time and we demonstrate how the training dataset $\mathcal{S}_t$ is updated to $\mathcal{S}_{t+1}$ in Fig. 1. Under such dynamics, both the ML system and the strategic population change over time and may lead to unexpected long-term consequences. An illustrating example is given in Fig. 2.

In this paper, we examine the evolution of the ML model and the agent data distribution. We ask: 1) How does the agent population evolve when the model is retrained with strategic feedback? 2) How is the ML system affected by the agent's strategic response? 3) If agents come from multiple social groups, how can model retraining further impact algorithmic fairness? Can imposing group fairness constraints during model training bring long-term societal benefits?

Compared to prior studies on *strategic classification* under sequential settings [5, 6, 7] that mainly focused on developing a classifier robust to strategic agents, we study the long-term impacts of model retraining with agent strategic feedback. Instead of assuming actual labels are available while retraining, we consider more practical scenarios with *model-annotated* samples. Although the risks on accuracy [3] and fairness [8] of using *model-annotated* samples to retrain models have been highlighted, ours is the first to incorporate strategic feedback from human agents. In App. C, we discuss more related works in detail. Our contributions are summarized as follows:

1. We formulate the problem of model retraining with human strategic feedback (Sec. 2).

2. We theoretically characterize the evolution of the expected *acceptance rate* (i.e., the proportion of agents receiving positive classifications), *qualification rate* (i.e., the proportion of agents with positive labels), and the *classifier bias* (i.e., the discrepancy between acceptance rate and qualification rate) under the retraining process. We show that the acceptance rate increases over time under retraining, while the actual qualification rate may decrease under certain conditions. The dynamics of *classifier bias* are more complex depending on the systematic bias of *human-annotated* samples. Finally, we propose an approach to stabilize the dynamics (Sec. 3).

3. We consider settings where agents come from multiple social groups and investigate how inter-group fairness can be affected by the model retraining process; we also investigate the long-term effects of fairness intervention at each round of model retraining (Sec. 4).
4. We conduct experiments on (semi-)synthetic and real data to verify the theorems (Sec. 5, App. E, App. F).

## 2    Problem Formulation

Consider a population of agents who are subject to certain ML decisions (e.g., admission/hiring decisions) and join the decision-making system in sequence. Each agent has observable continuous features $X \in \mathbb{R}^d$ and a hidden binary label $Y \in \{0, 1\}$ indicating its qualification state ("1" being qualified and "0" being unqualified). Let $P_{XY}$ be the joint distribution of $(X, Y)$ which is fixed over time, and $P_X$, $P_{Y|X}$ be the corresponding marginal and conditional distributions. Assume $P_X$, $P_{Y|X}$ are continuous with non-zero probability mass everywhere in their domain. For agents who join the system at time $t$, the decision-maker uses a classifier $f_t : \mathbb{R}^d \to \{0, 1\}$ to make decisions. Note that the decision-maker does not know $P_{XY}$ and can only learn $f_t$ from the training dataset at $t$ [9].

**Agent best response.** Agents who join the system at time $t$ can adapt their behaviors based on the latest classifier $f_{t-1}$ and change their features $X$ strategically. We denote the resulting data distribution as $P_{XY}^t$. Specifically, given original features $X = x$, agents have incentives to change their features at costs to receive positive classification outcomes, i.e., by maximizing utility

$$x_t = \arg\max_z \{f_{t-1}(z) - c(x, z)\} \tag{1}$$

where distance function $c(x, z) \geq 0$ measures the cost for an agent to change features from $x$ to $z$. In this paper, we consider $c(x, z) = (z - x)^T B(z - x)$ for some $d \times d$ positive semidefinite matrix $B$, allowing heterogeneous costs for different features. After agents best respond, their data distribution changes from $P_{XY}$ to $P_{XY}^t$. In this paper, we term $P_{XY}$ agent *prior-best-response* distribution and $P_{XY}^t$ *post-best-response* distribution. We consider natural settings that (i) agents need time to adapt their behaviors and their responses are *delayed* [10]: they act based on the latest classifier $f_{t-1}$ they are aware of, not the one they receive; (ii) agent behaviors are benign and feature changes can genuinely affect their underlying labels, so feature-label relationship $P_{Y|X}^t = P_{Y|X}$ is fixed over time [9, 11].

**Human-annotated samples and systematic bias.** At each round $t$, we assume the decision-maker can draw a limited number of unlabeled samples from the prior-best-response distribution $P_X$.[1] With some prior knowledge (possibly biased), the decision-maker can annotate these features and generate *human-annotated* samples $\mathcal{S}_{o,t}$. We assume the quality of human annotations is consistent, so $\mathcal{S}_{o,t}$ at any $t$ is drawn from a fixed probability distribution $D_{XY}^o$ with marginal distribution $D_X^o = P_X$. Because human annotations may not be the same as true labels, $D_{Y|X}^o$ can be biased compared to $P_{Y|X}$. We define such difference as the decision-maker's *systematic bias*, formally stated below.

**Definition 2.1** (Systematic bias). Define $\mu(D^o, P) := \mathbb{E}_{x \sim P_X}[D_{Y|X}^o(1|x) - P_{Y|X}(1|x)]$. The decision-maker has a systematic bias if $\mu(D^o, P) > 0$ (overestimation) or $< 0$ (underestimation).

Def. 2.1 implies that the decision-maker has a systematic bias when it labels a larger (or smaller) proportion of agents as qualified compared to the ground truth. In App. B, we present numerous examples of systematic bias in real applications where the decision-maker has different systematic biases towards different *demographic groups*. Generally, the systematic bias may or may not exist and we study both scenarios in the paper.

**Model-annotated samples.** In addition to human-annotated samples, the decision-maker at each round $t$ can also utilize the most recent classifier $f_{t-1}$ to generate *model-annotated* samples for training $f_t$. Specifically, let $\{x_{t-1}^i\}_{i=1}^N$ be $N$ post-best-response features ((1)) acquired from the

---

[1] We consider natural settings where the human-annotated samples are drawn from fixed $P_X$ and the process is independent of the decision-making process, i.e., the agents who best respond to $f_{t-1}$ are classified by model $f_t$ and the decision-maker never confuses them with human annotations. Instead, human annotation is a separate process for the decision-maker to obtain additional information about the whole population (e.g., by first acquiring data from public datasets or third parties, and then labeling them to estimate the population distribution $P_{XY}$). We provide an additional example and also consider the situation where *human-annotated* samples are drawn from *post-best-response* distribution $P_X^t$ in App. D.1.

agents coming at $t-1$, the decision-maker uses $f_{t-1}$ to annotate the samples and obtain *model-annotated* samples $\mathcal{S}_{m,t-1} = \{x^i_{t-1}, f_{t-1}(x^i_{t-1})\}^N_{i=1}$. Both human and model-annotated samples are used to retrain the classifier at $t$.

**Classifier's retraining process.** With the human and model-annotated samples, we next introduce how the model is retrained by the decision-maker over time. Denote the training dataset at $t$ as $\mathcal{S}_t$. Initially, the decision-maker trains $f_0$ with a *human-annotated* training dataset $\mathcal{S}_0 = \mathcal{S}_{o,0}$. Then the decision-maker updates $f_t$ every round to make decisions about agents, and it learns $f_t$ using empirical risk minimization (ERM) with training dataset $\mathcal{S}_t$. Similar to studies in strategic classification [2], we consider linear classifier in the form of $f_t(x) = \mathbf{1}(h_t(x) \geq \theta)$ where $h_t : \mathbb{R} \to [0,1]$ is the scoring function (e.g., logistic function) and $h_t \in \mathcal{H}$. At each round $t \geq 1$, $\mathcal{S}_t$ consists of three components: existing training samples $\mathcal{S}_{t-1}$, $N$ new *model-annotated* and $K$ new *human-annotated* samples:

$$\mathcal{S}_t = \mathcal{S}_{t-1} \cup \mathcal{S}_{m,t-1} \cup \mathcal{S}_{o,t-1}, \ \ \forall t \geq 1 \tag{2}$$

Since annotating agents is usually time-consuming and expensive, we have $N \gg K$ in practice. The complete retraining process is shown in Alg. 1 (App. A).

Given the post-best-response distribution $P^t_{XY}$, we can define the associated ***qualification rate*** as the probability that agents are qualified, i.e.,

$$Q(P^t) = \mathbb{E}_{(x,y) \sim P^t_{XY}} [y].$$

For the classifier $f_t$ deployed on marginal feature distribution $P^t_X$, we define ***acceptance rate*** as the probability that agents are classified as positive, i.e.,

$$A(f_t, P^t) = \mathbb{E}_{x \sim P^t_X}[f_t(x)].$$

Since $\mathcal{S}_t$ is randomly sampled at all $t$, the resulting classifier $f_t$ and agent best response are also random. Denote $D^t_{XY}$ as the probability distribution of sampling from $\mathcal{S}_t$ and recall that $D^o_{XY}$ is the distribution for *human-annotated* $\mathcal{S}_{o,t}$, we can further define the expectations of $Q(P^t), A(f_t, P^t)$ over the training dataset:

$$q_t := \mathbb{E}_{\mathcal{S}_{t-1}}[Q(P^t)]; \quad a_t := \mathbb{E}_{\mathcal{S}_t}[A(f_t, P^t)],$$

where $q_t$ is the expected actual qualification rate of agents after they best respond, note that the expectation is taken with respect to $\mathcal{S}_{t-1}$ because the distribution $P^t_{XY}$ is the result of agents responding to $f_{t-1}$ which is trained with $\mathcal{S}_{t-1}$; $a_t$ is the expected acceptance rate of agents at time $t$.

**Dynamics of qualification rate & acceptance rate.** Under the model retraining process, both the model $f_t$ and agent distribution $P^t_{XY}$ change over time. One goal is to understand how the agents and the ML model interact and impact each other in the long run. Specifically, we are interested in the dynamics of the following variables:

1. **Qualification rate** $q_t$: it measures the qualification of agents and indicates the *social welfare*.
2. **Acceptance rate** $a_t$: it measures the likelihood that an agent can receive positive outcomes and indicates the *applicant welfare*.
3. **Classifier bias** $\Delta_t = |a_t - q_t|$: it is the discrepancy between the acceptance rate and the true qualification rate, measuring how well the decision-maker can approximate agents' actual qualification rate and can be interpreted as *decision-maker welfare*.

In the rest of the paper, we study the dynamics of $q_t, a_t, \Delta_t$ and we aim to answer the following questions: 1) How do the qualification rate $q_t$, acceptance rate $a_t$, and classifier bias $\Delta_t$ evolve under the dynamics? 2) How can the evolution of the system be affected by the decision-maker's retraining process? 3) What are the impacts of the decision-maker's systematic bias? 4) If we further consider agents from multiple social groups, how can the retraining process affect inter-group fairness?

## 3  Dynamics of the Agents and Model

In this section, we examine the evolution of qualification rate $q_t$, acceptance rate $a_t$, and classifier bias $\Delta_t$. We aim to understand how *applicant welfare* (Sec. 3.1), *social welfare* (Sec. 3.2), and *decision-maker welfare* (Sec. 3.3) are affected by the retraining process in the long run. We first introduce some assumptions used for the theorems.

**Assumption 3.1.** Hypothesis class $\mathcal{H}$ can perfectly learn the training data distribution $D_{Y|X}^t$, i.e., $\exists h_t^* \in \mathcal{H}$ such that $h_t^*(x) = D_{Y|X}^t(1|x)$.

With Assumption 3.1, we avoid the effects of learning error on the system dynamics; this allows us to focus on the dynamic interactions between strategic agents and ML system. Although the theoretical analysis relies on the assumption, our experiments in Sec. 5 and App. F show consistent results with theorems. We further assume the monotone likelihood ratio property holds for $D_{XY}^o$ and $P_{XY}$.

**Assumption 3.2.** Let $x[m]$ be the $m^{th}$ dimension of $x \in \mathbb{R}^d$, then $D_{Y|X}^o(1|x)$ and $P_{Y|X}(1|x)$ are continuous and monotonically increasing in $x[m]$, $\forall m = 1, \cdots, d$ while other dimensions are fixed.

Note that the Assumption 3.2 is mild and widely used in previous literature [12, 13]. It can be satisfied by many distributional families such as exponential, Gaussian, and mixtures of exponential/Gaussian. It implies that agents are more likely to be qualified as feature value increases.

### 3.1 Applicant welfare: dynamics of acceptance rate

We first examine the dynamics of $a_t = \mathbb{E}_{\mathcal{S}_t}[A(f_t, P^t)]$. Intuitively, under Assumption 3.1, all classifiers can fit the training data well. Then the model-annotated samples $\mathcal{S}_{m,t-1}$ generated from post-best-response agents would have a higher qualification rate than the qualification rate of training data $\mathcal{S}_{t-1}$. As a result, the training data $\mathcal{S}_t$ augmented with $\mathcal{S}_{m,t-1}$ has a higher proportion of qualified agents than the qualification rate of $\mathcal{S}_{t-1}$, thereby producing a more "generous" classifier $f_t$ with a larger $a_t$. This reinforcing process can be formally stated in Thm. 3.3.

**Theorem 3.3** (Evolution of $a_t$). *Under the retraining process, the acceptance rate of the agents that join the system increases over time, i.e., $a_t > a_{t-1}$, $\forall t \geq 1$.*

We prove Thm. 3.3 by mathematical induction in App. G.3 and Fig. 3 illustrates the theorem. When agents best respond, the decision-maker tends to accept more agents. We can further show that when the number of *model-annotated* samples $N$ is large compared to the number of *human-annotated* samples $K$, the classifier will ultimately accept all agents in the long run (Prop. 3.4).

**Proposition 3.4.** *For any $P_{XY}, D^o, B$, there exists a threshold $\lambda > 0$ such that $\lim_{t \to \infty} a_t = 1$ whenever $\frac{K}{N} < \lambda$.*

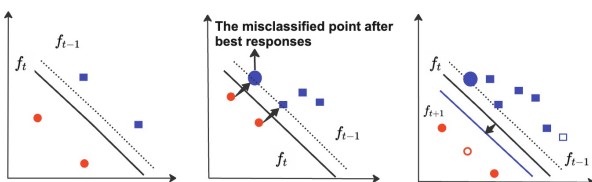

Figure 3: Increasing acceptance rate from $a_t$ to $a_{t+1}$. Unqualified/qualified agents are shown as circles/squares, while the admitted/rejected agents are shown in red/blue. New agents coming at $t+1$ are shown in hollow. **The left plot** shows the training set $\mathcal{S}_t$ containing 2 unqualified (red circle) and 2 qualified agents (blue square) and $a_t$ is 0.5. **The middle plot** shows the agents coming at $t$ best respond to $f_{t-1}$. After the responses, 3 of 4 agents are qualified (blue square) and 1 is still unqualified (blue circle). However, all 4 agents are annotated as "qualified" (blue). **The right plot** shows the training set $\mathcal{S}_{t+1}$ containing all points of the left and middle plot, plus two new human-annotated points (hollow points). All blue points are labeled as 1 and the red points are 0. So $\mathcal{S}_{t+1}$ has more samples with a positive label (0.7), resulting in $f_{t+1}$ accepting a higher proportion of agents.

The specific value of $\lambda$ in Prop. 3.4 depends on $P_{XY}, D^o, B$, which is difficult to find analytically. Nonetheless, we illustrate in Sec. 5 that when $\frac{K}{N} = 0.05$, $a_t$ tends to approach 1 in numerous datasets. Since the *human-annotated* samples are often difficult to attain (due to time and labeling costs), the condition in Prop. 3.4 is easy to satisfy in practice.

### 3.2 Social welfare: dynamics of qualification rate

Next, we study the dynamics of qualification rate $q_t = \mathbb{E}_{\mathcal{S}_{t-1}}[Q(P^t)]$. Unlike the acceptance rate $a_t$ which always increases during the retraining process, the evolution of $q_t$ is more complicated and depends on agent prior-best-response distribution $P_{XY}$.

Specifically, let $q_0 = Q(P) = \mathbb{E}_{(x,y) \sim P_{XY}}[y]$ be the initial qualification rate, then the difference between $q_t$ and $q_0$ can be interpreted as the amount of *improvement* (i.e., increase in label) agents

gain from their strategic behavior at $t$. This is determined by (i) the proportion of agents that decide to change their features at costs (depends on $P_X$), and (ii) the improvement agents can expect upon changing features (depends on $P_{Y|X}$). Thus, the dynamics of $q_t$ depend on $P_{XY}$. Despite the intricate nature of dynamics, we can still derive a condition under which $q_t$ decreases monotonically.

**Theorem 3.5** (Evolution of $q_t$). *Consider the setting where the $d$-dimensional feature space $X \in \mathbb{R}^d$ where $F_X(x), P_X(x), P_{Y|X}(1|x)$ are the cumulative distribution function, probability density function and the labeling function when $Y = 1$. Denote $\mathcal{J} = \{x|f_0(x) = 0\}$ as the half-space in $\mathbb{R}^d$ determined by the classifier $f_0$. Under the retraining process, $\forall t \geq 1$, $q_{t+1} \leq q_t$ if either of the following conditions holds: (i) $F_X$ and $P_{Y|X}(1|x)$ are convex on $\mathcal{J}$; (ii) for each dimension $x[i], i \in [d]$, $F_X(x)$ and $P_{Y|X}(1|x)$ are convex with respect to $x[i]$.*

Note that $q_{t+1} \leq q_t$ in Thm. 3.5 holds only for $t \geq 1$. Because agent behavior can only improve their labels, prior-best-response $q_0$ always serves as the lower bound of $q_t$. The half-space $\mathcal{J}$ in Thm. 3.5 specifies the region in feature space where agents have incentives to change their features. The convexity of $F_X$ and $P_{Y|X}(1|x)$ ensure that as $f_t$ evolves from $t = 1$: (i) fewer agents choose to improve their features, and (ii) agents expect less improvement from feature changes. Thus, $q_t$ decreases over time. Conditions in Thm. 3.5 can be satisfied by common distributions $P_X$ (e.g., Uniform, Beta($\alpha, 1$) with $\alpha > 1$) and labeling functions $P_{Y|X}(1|x)$ (e.g., linear function, quadratic functions with degree greater than 1). The proof and a more general analysis are shown in App. G.5. We also show that Thm. 3.5 is valid under diverse experimental settings (Sec. 5, App. E, App. F).

### 3.3 Decision-maker welfare: dynamics of classifier bias

Sec. 3.1 and 3.2 show that as the classifier $f_t$ gets updated over time, agents are more likely to get accepted ($a_t$ increases). However, their true qualification rate $q_t$ (after the best response) may actually decrease. It indicates that the decision-maker's misperception about agents varies over time. Thus, this section studies the dynamics of classifier bias $\Delta_t = |a_t - q_t|$. Our results show that the evolution of $\Delta_t$ is largely affected by the decision-maker's systematic bias $\mu(D^o, P)$ as defined in Def. 2.1.

**Theorem 3.6** (Evolution of $\Delta_t$). *Starting from $t = 1$ at the retraining process and under conditions in Thm. 3.5:*

1. *If $\mu(D^o, P) = 0$, i.e., the systematic bias does not exist, then $\Delta_t$ increases over time.*

2. *If $\mu(D^o, P) > 0$, i.e., the decision-maker overestimates agent qualification, then $\Delta_t$ increases over time.*

3. *If $\mu(D^o, P) < 0$, i.e., the decision-maker underestimates agent qualification, $\Delta_t$ **either** monotonically decreases **or** first decreases but then increases.*

Thm. 3.6 highlights the potential risks of the model retraining process and is proved in App. G.6. Originally, the purpose of retraining the classifier was to ensure accurate decisions on the targeted population. However, when agents behave strategically, the retraining may lead to adverse outcomes by amplifying the classifier bias. Meanwhile, though systematic bias is usually an undesirable factor to eliminate when learning ML models, it may help mitigate classifier bias to improve the *decision-maker welfare* in the retraining process, i.e., $\Delta_t$ decreases when $\mu(D^o, P) < 0$.

### 3.4 Intervention to stabilize the dynamics

Sec. 3.1- 3.3 show that as the model is retrained from strategic agents, $a_t, q_t, \Delta_t$ are unstable and may change monotonically over time. Next, we introduce an effective approach to stabilizing the system.

From the above analysis, we know that one reason that makes $q_t, a_t, \Delta_t$ evolve is agent's best response, i.e., agents improve their features strategically to be accepted by the most recent model, which leads to a higher qualification rate of *model-annotated* samples (and the resulting training data), eventually causing $a_t$ to deviate from $q_t$. Thus, to mitigate such deviation, we can improve the quality of model annotation. Our method is proposed based on this idea, which uses a *probabilistic sampler* [3] when producing *model-annotated* samples.

Specifically, at each time $t$, instead of adding $\mathcal{S}_{m-1,o} = \{x_{t-1}^i, f_{t-1}(x_{t-1}^i)\}_{i=1}^N$ (samples annotated by the model $f_{t-1}$ to training data $\mathcal{S}_t$ (2)), we use the probabilistic model $h_{t-1}(x)$ to annotate each sample according to the following: *For each sample $x$, we label it as 1 with probability $h_{t-1}(x)$, and as 0 otherwise.* Here $h_{t-1}(x) \approx D_{Y|X}^{t-1}(1|x)$ is the estimated posterior probability learned from $\mathcal{S}_{t-1}$

(e.g., using logistic model). We call the procedure **refined retraining process** if *model-annotated* samples are generated in this way based on a probabilistic sampler.

Fig. 3 also illustrates the above idea: agents best respond to $f_{t-1}$ (middle plot) to improve and $f_t$ will label both as $1$. By contrast, a probabilistic sampler $h_t$ only labels a fraction of them as $1$. This alleviates the influence of agents' best responses to stabilize the dynamics of $a_t, q_t, \Delta_t$. Prop. D.2 and App. F.3 provide proofs and more experiments for the **refined retraining process**.

## 4 Impacts on Algorithmic Fairness

In this section, we further consider agents from multiple social groups and investigate how group fairness can be affected by the model retraining process. Similar to prior studies in fair ML [12, 14, 15], we assume the decision-maker knows the group identity of each agent and uses group-dependent classifiers to make decisions. WLOG, we present the results for any pair of two groups $i, j$. Among the two groups, we define the group with a smaller acceptance rate under unconstrained optimal classifiers as **disadvantaged group**.

### 4.1 Impacts of systematic bias & model retraining

We first consider the situation where two groups have no innate difference: they have the same prior-best-response feature distribution and the same cost matrix $B$ to change features. However, the decision-maker has a systematic bias in favor of group $i$ more than group $j$, making $i$ the advantaged group and $j$ the disadvantaged group. We consider the fairness metric *demographic parity* (DP) [16], which measures unfairness as the difference in acceptance rate across two groups. Extension to other fairness metrics such as *equal opportunity* [17] is discussed in App. D.2. Thm. 4.1 below shows the long-term impacts of refined retraining process (i.e., model-annotated samples generated with probabilistic sampler $h_t$) and original model retraining process (i.e., model-annotated samples generated with model $f_t$) on group unfairness.

**Theorem 4.1** (Impacts of model retraining on unfairness). *When groups $i, j$ have no innate difference, but $j$ is disadvantaged due to systematic bias: (i) If applying **original retraining process to both groups** with $\frac{K}{N}$ satisfying Prop. 3.4, then group $j$ will stay disadvantaged until all agents in both groups are accepted to achieve perfect fairness; (ii) If applying **refined retraining process to both groups**, then group $j$ will stay disadvantaged and the unfairness remains the same in the long run; (iii) If applying **refined retraining process to group $i$ but the original process to group $j$** with $\frac{K}{N}$ satisfying Prop. 3.4, then unfairness first decreases after certain rounds of retraining until group $j$ becomes advantaged, then unfairness increases.*

Thm. 4.1 shows that original and refined retraining processes impact differently on group fairness as proved in App. G.9. Specifically, the original retraining process ultimately attains "trivial" perfect fairness by accepting all agents, whereas the refined retraining process stabilizes the dynamics but unfairness always exists. Interestingly, applying disparate retraining strategies to two groups may result in perfect fairness in the middle of retraining process.

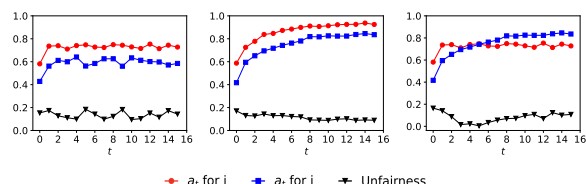

Figure 4: Unfairness (DP) when the refined retraining process is applied to both groups (left), original retraining process is applied to both groups (middle), refined retraining process is only applied to group $i$ (right) under dataset 2.

This suggests that it may be beneficial for the decision-maker to monitor the fairness measure during the retraining and execute an *early stopping* mechanism to attain almost perfect DP fairness, i.e., stop retraining models early once unfairness reaches the minimum. As shown in the right plot of Fig. 4, when refined retraining is only applied to group $i$, unfairness is minimized at $t = 5$.

### 4.2 Impact of short-term fairness intervention

Sec. 4.1 proposed a solution of "disparate retraining with early stopping" to mitigate unfairness. A more common method to maintain fairness throughout the retraining process is to enforce certain fairness constraints every time when updating the models. Next, we consider this method where

*refined retraining process* is applied to both groups (to stabilize dynamics) and a fairness constraint is imposed at each round of model retraining. Unlike Sec. 4.1, we consider a general setting where two groups may have different feature distributions and different cost matrices $B$, but feature-label relation $P_{Y|X}(1|x)$ is the same across groups.

**Finding fair models.** For each group $s$, we use superscript $s$ to denote the group-specific distributions/metrics listed in Sec. 2 (e.g., $P_{XY}^{s,t}$). At each round $t$, the decision-maker first trains a linear model $f_t^s = \mathbf{1}(h_t^s(x) \geq \theta)$ for group $s$. According to Assumption 3.1 and Prop. D.2, $h_t^s$ is expected to be $D_{Y|X}^{o,s}(1|x) = P_{Y|X}^s(1|x) + \mu_s$, where $\mu_s$ is the systematic bias towards group $s$. Denote $\theta_t^s$ as the original optimal threshold at $t$ without the fairness intervention, and let $a_t^s$ be the acceptance rate for group $s$ under $\theta_t^s$, then the decision-maker can tune the thresholds to get fair-optimal thresholds $\widetilde{\theta}_t^s$ (a pair of thresholds satisfying the fairness constraint with the largest aggregated accuracy).

**Noisy agent best response.** To ensure there always exists a pair of thresholds that satisfy DP fairness constraint (i.e., equal acceptance rate), each group's post-best-response distribution needs to be continuous. However, when agents modify their features based on (1), the aggregate response necessarily exhibits discontinuities [18]. To tackle this issue, we consider the *noisy best response* model proposed by Jagadeesan et al. [18], which assumes the agents only have imperfect and noisy information of decision threshold. Formally, given decision threshold $\theta$, each agent best responds to $\theta + \epsilon$ where $\epsilon$ is a noise independently sampled from a zero mean distribution with finite variance $\sigma^2$. Under noisy best response, we investigate the impacts of fairness intervention.

**Theorem 4.2** (Impact of fairness intervention). *Suppose group $j$ is disadvantaged from round $0$ to $t$, i.e., group $j$ has a smaller acceptance rate than group $i$ under unconstrained optimal thresholds $\{\theta_\tau^i, \theta_\tau^j\}, \forall \tau \leq t$. Let $\sigma_t^2$ be the variance of the noisy best response at $t$. We have the following:*

*(i) **Without** fairness intervention, $\forall \sigma_t$, there always exists feature distributions and cost matrices under which group $j$ switches to be advantaged at $t + 1$;*

*(ii) **With** fairness intervention, if $\sigma_t < (\theta_t^j - \widetilde{\theta}_t^j)\sqrt{a_0^i - a_t^j}$, then group $j$ always remains disadvantaged at $t + 1$.*

Thm. 4.2 shows that without fairness intervention, the originally disadvantaged group $j$ can flip to be advantaged. In contrast, the fairness intervention helps maintain the disadvantaged and advantaged groups when the agent's perception of the decision rule is sufficiently accurate. Note that the bound on $\sigma$ is well-defined. Since group $j$ is disadvantaged, $a_0^i > a_0^j$ always holds. If $\sigma_t < (\theta_t^j - \widetilde{\theta}_t^j)\sqrt{a_0^i - a_t^j}$ holds, then it is guaranteed that $a_0^i > a_{t+1}^j$ (see App. G.10 for details). This result implies that the disadvantaged group, by losing their chance to become advantaged, may not benefit from fairness intervention in the long run.

## 5 Experiments

We conduct experiments on two synthetic (Uniform, Gaussian), one semi-synthetic (German Credit [19]), and one real dataset (Credit Approval [20]) to validate the dynamics of $a_t, q_t, \Delta_t$ and the unfairness [2]. Note that only the Uniform dataset satisfies all assumptions and the conditions in our theoretical analysis, while the Gaussian and German Credit datasets violate the conditions in Thm. 3.5. The Credit Approval dataset violates all assumptions and conditions of the main paper. The decision-maker trains logistic regression models for all experiments using stochastic gradient descent (SGD) over $T$ steps. We present the experimental results of the Gaussian and German Credit datasets to illustrate the dynamics of $a_t, q_t, \Delta_t$ in this section, while the results for Uniform and Credit Approval data are similar and shown in App. E.

**Gaussian data.** We consider a synthetic dataset with Gaussian distributed $P_X$. $P_{Y|X}$ is logistic and satisfies Assumption 3.2 but not the conditions of Thm. 3.5. We assume agents have two independent features $X_1, X_2$ and are from two groups $i, j$ with

Table 1: Gaussian Dataset Setting

| $P_{X_k}(x_k)$ | $P_{Y|X}(1|x)$ | $n, r, T, q_0$ |
|---|---|---|
| $\mathcal{N}(0, 0.5^2)$ | $(1 + \exp(-x_1 - x_2))^{-1}$ | $100, 0.05, 15, 0.5$ |

different sensitive attributes but identical joint distribution $P_{XY}$. Their cost matrix is $B = \begin{bmatrix} 5 & 0 \\ 0 & 5 \end{bmatrix}$

[2]https://github.com/osu-srml/Automating-Data-Annotation-under-Strategic-Human-Agents

and the initial qualification rate is $q_0 = 0.5$. We assume the decision-maker has a systematic bias by overestimating (resp. underestimating) the qualification of agents in the advantaged group $i$ (resp. disadvantaged group $j$), which is modeled as increasing $D_{Y|X}^o(1|x)$ to be 0.1 larger (resp. smaller) than $P_{Y|X}(1|x)$ for group $i$ (resp. group $j$). For the retraining process, we let $r = \frac{K}{N} = 0.05$ (i.e., the number of model-annotated samples $N = 2000$, which is sufficiently large compared to the number of human-annotated samples $K = 100$). Table 1 summarizes the dataset information, and the joint distributions are visualized in App. F.1.

We verify the results in Sec. 3 by illustrating the dynamics of $a_t, q_t, \Delta_t$ for both groups (Fig. 5a). Since our evolution results are in expectation, we perform $n = 100$ independent runs of experiments for every parameter configuration and show the averaged outcomes. The results are consistent with Thm. 3.3, 3.5 and 3.6: (i) acceptance rate $a_t$ (red curves) increases monotonically; (ii) qualification rate $q_t$ decreases monotonically starting from $t = 1$ (since strategic agents only best respond from $t = 1$); (iii) classifier bias $\Delta_t$ evolves differently for different groups and it may reach the minimum after a few rounds of retraining.

We further test the robustness of system dynamics against agent noisy response, where we assume agents estimate their outcomes as $\hat{f}_t(x) = f_t(x) + \epsilon$ with $\epsilon \sim \mathcal{N}(0, 0.1)$. We present the dynamics of $a_t, q_t, \Delta_t$ for both groups in Fig. 6a which are similar to Fig. 5a, demonstrating the robustness of our theorems.

**German Credit dataset [19].** This dataset includes features for predicting individuals' credit risks. It has 1000 samples and 19 numeric features, which are used to construct a larger-scaled dataset. Specifically, we fit a kernel density estimator for all 19 features to generate 19-dimensional features, the corresponding labels are sampled from the distribution $P_{Y|X}$ which is estimated from data by fitting a logistic classifier with 19 features. Given this dataset, the first 10 features are used to train the classifiers. The attribute "sex" is regarded as the sensitive attribute. The systematic bias is created by increasing/decreasing $P_{Y|X}$ by 0.06. Other parameters $n, r, T, q_0$ are the same as Table 1. Since $P_{Y|X}$ is a logistic function, Assumption 3.2 can be satisfied easily as illustrated in App. F.1.

We verify the results in Sec. 3 by illustrating the dynamics of $a_t, q_t, \Delta_t$ for both groups (Fig. 5b). The results are consistent with Thm. 3.3, 3.5 and 3.6: (i) acceptance rate $a_t$ (red curves) always increases; (ii) qualification rate $q_t$ (blue curves) decreases starting from $t = 1$ (since strategic agents only best respond from $t = 1$); (iii) classifier bias $\Delta_t$ (black curves) evolve differently for different groups. Finally, similar to Fig. 5b, Fig. 6b demonstrates the results are still robust under the noisy setting.

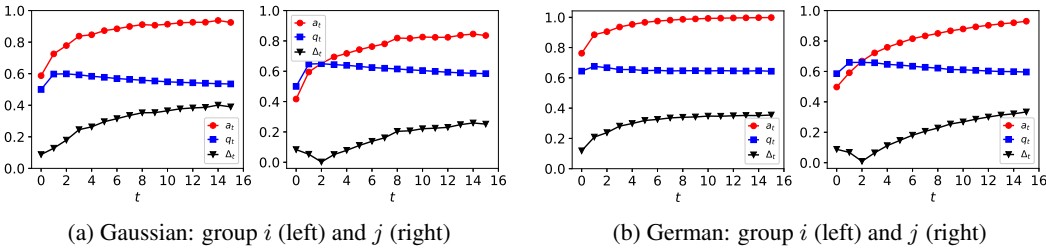

(a) Gaussian: group $i$ (left) and $j$ (right)  (b) German: group $i$ (left) and $j$ (right)

Figure 5: Dynamics of $a_t, q_t, \Delta_t$ under the perfect information setting

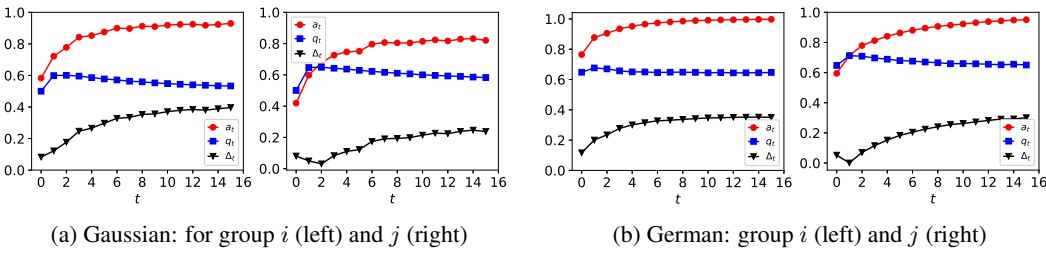

(a) Gaussian: for group $i$ (left) and $j$ (right)  (b) German: group $i$ (left) and $j$ (right)

Figure 6: Dynamics of $a_t, q_t, \Delta_t$ under the noisy setting

**Additional experiments.** We provide more results in App. F to: (i) verify Thm. 3.3 and Thm. 3.6; (ii) Visualize the influence of **each factor** including training rounds, cost matrices, the ratio between *human-annotated* and *model-annotated* samples, whether the agents are strategic, and whether certain

assumptions are violated; (iii) visualize the evolution of unfairness when different retraining strategies are applied to different groups.

# 6 Conclusion & Limitations

This paper studies the dynamics where strategic agents interact with an ML system retrained over time with *model-annotated* and *human-annotated* samples. We rigorously studied the evolution of *applicant welfare*, *decision-maker welfare*, and *social welfare*. Such results highlight the potential risks of retraining classifiers when agents are strategic. The paper also provides a comprehensive analysis on the fairness dynamics associated with the retraining process, revealing that the fairness intervention may not bring long-term benefits. To ease the negative social impacts, we provide mechanisms to stabilize the dynamics and an early stopping mechanism to maintain fairness. However, our theoretical results rely on certain assumptions and we should first verify these conditions before adopting the results of this paper, which may be challenging in real-world applications.

## Acknowledgement

This material is based upon work supported by the U.S. National Science Foundation under award IIS-2202699 and IIS-2416895, by OSU President's Research Excellence Accelerator Grant, and grants from the Ohio State University's Translational Data Analytics Institute and College of Engineering Strategic Research Initiative.

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

# A The retraining process

---

**Algorithm 1** retraining process

---

**Input:** Joint distribution $D^o_{XY}$ for any $\mathcal{S}_{o,t}$, Hypothesis class $\mathcal{F}$, the number of the initial training samples and agents coming per round $N$, the number of decision-maker-labeled samples per round $K$.

**Output:** Model deployments over time $f_0, f_1, f_2, \ldots$
1: $S_0 = S_{o,0} \sim D^o_{XY} = \{x^i_0\}^N_{i=1}$
2: At $t = 0$, deploy $f_0 \sim \mathcal{F}(\mathcal{S}_0)$
3: **for** $t \in \{1, \ldots \infty\}$ **do**
4: $\quad$ $N$ agents gain knowledge of $f_{t-1}$ and best respond to it, resulting in $\{x^i_t\}^N_{i=1}$
5: $\quad$ $\mathcal{S}_{m,t} = \{x^i_{t-1}, f_{t-1}(x^i_{t-1})\}^N_{i=1}$ consists of the model-labeled samples from round $t - 1$.
6: $\quad$ $\mathcal{S}_{o,t} \sim D^o_{XY}$ consists of the new $K$ decision-maker-labeled samples.
7: $\quad$ $\mathcal{S}_t = \mathcal{S}_{t-1} \cup \mathcal{S}_{m,t} \cup \mathcal{S}_{o,t}$
8: $\quad$ Deploy $f_t \sim \mathcal{F}(\mathcal{S}_t)$ on the incoming $N$ agents who best respond to $f_{t-1}$ with the resulting joint distribution $P^t_{XY}$.
9: **end for**

---

# B Motivating examples of the systematic bias

Def. 2.1 highlights the systematic nature of the decision-maker's bias. This bias is quite ubiquitous when labeling is not a trivial task. It almost always the case when the decision-maker needs to make human-related decisions. We provide the following motivating examples of systematic bias with supporting literature in social science:

1. *College admissions*: consider experts in the admission committee of a college that obtains a set of student data and wants to label all students as "qualified" or "unqualified". The labeling task is much more complex and subjective than the ones in computer vision/natural language processing which have some "correct" answers. Therefore, the experts in the committee are prone to bring their "biases" towards a specific population sharing the same sensitive attribute into the labeling process including:

   (a) *Implicit bias:* the experts may have an implicit bias they are unaware of to favor/discriminate against students from certain groups. For instance, a famous study [21] reveals admission committee members at the medical school of the Ohio State University unconsciously have a "better impression" towards white students; Alvero et al. [22] finds out that even when members in an admission committee do not access the sensitive attributes of students, they unconsciously infer them and discriminate against students from the minority group.

   (b) *Selection bias:* the experts may have insufficient knowledge of the under-represented population due to the selection bias [23] because only a small portion of them were admitted before. Thus, experts may expect a lower qualification rate from this population, resulting in more conservative labeling practices. The historical stereotypes created by selection bias are difficult to erase.

2. *Loan applications*: consider experts in a big bank that obtains data samples from some potential applicants and wants to label them as "qualified" or "unqualified". Similarly, the experts are likely to have systematic bias including:

   (a) *Implicit bias:* similarly, Brock and De Haas [24] conduct a lab-in-the-field experiment with over 300 Turkish loan officers to show that they bias against female applicants even if they have identical profiles as male applicants.

   (b) *Selection bias:* when fewer female applicants are approved historically, the experts have less knowledge on females (i.e., whether they will actually default or repay), thereby tending to stay conservative.

# C   Related Work

## C.1   Strategic Classification

**Strategic classification without label changes.** Our work is mainly based on an extensive line of literature on strategic classification [1, 2, 5, 6, 7, 12, 14, 18, 25, 26, 27, 28, 29, 30, 31, 32]. These works assume the agents are able to best respond to the policies of the decision-maker to maximize their utilities. Most works modeled the strategic interactions between agents and the decision-maker as a repeated Stackelberg game where the decision-maker leads by publishing a classifier and the agents immediately best respond to it. The earliest line of works focused on the performance of regular linear classifiers when strategic behaviors never incur label changes [1, 5, 7, 25], while the later literature added noise to the agents' best responses [18], randomized the classifiers [26] and limited the knowledge of the decision-maker [28]. Levanon and Rosenfeld [2] proposed a generalized framework for strategic classification and a *strategic hinge loss* to better train strategic classifiers, but the strategic behaviors are still not assumed to cause label changes.

**Strategic classification with label changes.** Several other lines of literature enable strategic behaviors to cause label changes. The first line of literature mainly focuses on incentivizing improvement actions where agents have budgets to invest in different actions and only some of them cause the label change (improvement) [11, 13, 33, 34, 35, 36, 37, 38, 39, 40, 41]. The other line of literature focuses on *causal strategic learning* [32, 42, 43, 44, 45]. These works argue that every strategic learning problem has a non-trivial causal structure which can be explained by a *structural causal model*, where intervening on causal nodes causes improvement and intervening on non-causal nodes means manipulation.

**Performative prediction.** Several works consider *performative prediction* as a more general setting where the feature distribution of agents is a function of the classifier parameters. Perdomo et al. [46] first formulated the prediction problem and provided iterative algorithms to find the stable points of the model parameters. Izzo et al. [27] modified the gradient-based methods and proposed the `Perfgrad` algorithm. Hardt et al. [47] elaborated the model by proposing *performative power*.

**Retraining under strategic settings**

Most works on learning algorithms under strategic settings consider developing robust algorithms that the decision-maker only trains the classifier once [1, 2, 18, 28], while *Performative prediction*[46] focuses on developing online learning algorithms for strategic agents. There are only a few works [32, 48] which permit retraining the Strategic classification models. However, all these algorithms assume that the decision-maker has access to a new training dataset containing both agents' features and labels at each round. There is no work considering the dynamics under the retraining process with *model-annotated* samples. Also, few works [14, 15] discussed the long-term fairness issues under sequential strategic settings. Liu et al. [15] modeled strategic behaviors in a completely different way where each individual decides whether or not (a binary choice) to acquire the desired qualification based on a cost drawn from a fixed distribution. Moreover, they assumed the decision maker has perfect knowledge of the agent distribution, i.e., can draw infinitely many examples from the agent population. Instead, our work focuses on a practical setting where the decision maker must learn from samples and acquiring human-annotated samples is quite expensive, motivating the use of model-annotated samples as well. Zhang et al. [14] also simplified the modeling of the qualification changes of agents by using fixed transition matrices.

## C.2   Bias amplification during retraining

There has been an extensive line of study on the computer vision field about how machine learning models amplify the dataset bias, while most works only focus on the one-shot setting where the machine learning model itself amplifies the bias between different groups in one training/testing round [49, 50]; Another work theoretically studies *dynamic benchmarking* [51] to propose a retraining scheme to increase model accuracy with adversarial human-annotation. In recent years, another line of research focuses on the amplification of dataset bias under *model-annotated data* where ML models label new samples on their own and add them back to retrain themselves[4, 52, 53, 54, 55, 56, 57, 58, 59]. These works study the bias amplification in different practical fields including resource allocation [57], computer vision [54], natural language processing [55], generative models [60, 61] and clinical trials [59]. The most related work is [3] which studied the influence of retraining in the

non-strategic setting. Also, there is a work [4] touching on the data feedback loop under performative setting, but it focused on empirical experiments under medical settings where the feature distribution shifts are mainly caused by treatment and the true labels in historical data are highly accessible. Besides, the data feedback loop is also related to recommendation systems. where extensive works have studied how the system can shape users' preferences and disengage the minority population [58, 62, 63, 64]. However, previous literature did not touch on the retraining process.

### C.3 Machine learning fairness in strategic classification

Several works have considered how different fairness metrics [1, 9, 16, 65] are influenced in strategic classification [12, 14, 15, 66]. The most related works [12, 15] studied how strategic behaviors and the decision-maker's awareness can shape long-term fairness. They deviated from our paper since they never considered retraining and the strategic behaviors never incurred label changes.

## D  Additional Discussions

### D.1  The source of Human-annotated samples

As stated in footnote 1, for the source of human-annotated samples, we consider natural settings where the human-annotated samples are drawn from fixed $P_X$ and the process is independent of the decision-making process, i.e., the agents who best respond to $f_{t-1}$ are classified by model $f_t$ and the decision-maker never confuses them with human annotations. Instead, human annotation is a separate process for the decision-maker to obtain additional information about the whole population (e.g., by first acquiring data from public datasets or third parties, and then labeling them to estimate the population distribution $P_{XY}$). Here we provide an additional **example**: In a university admission scenario, the admission office uses the model $f_t$ to assign decisions at $t$ and obtain model-annotated samples for $t + 1$. But for human-annotated samples, the admission committee hopes to acquire a more objective knowledge of "what kinds of students are successful" from the whole student population. Thus, in the hope of avoiding additional sampling/selection bias, the committee seeks human annotations from third-party education researchers/experts. The experts can provide qualifications of the public/historical student samples drawn from the whole student population including students who are not applicants for this specific university or even do not attend a university.

In this section, we additionally consider the situation where all *human-annotated* samples at $t$ are drawn from the *post-best-response* distribution $P_X^t$. This will change (4) to the following:

$$\overline{q'}_t = \frac{tN+(t-1)K}{(t+1)N+tK} \cdot \overline{q'}_{t-1} + \frac{N}{(t+1)N+tK} \cdot a'_{t-1} + \frac{K}{(t+1)N+tK} \cdot q^*_{t-1} \tag{3}$$

where $q'$ and $a'$ denote the new qualification rate and acceptance rate, and $q^*_{t-1}$ stands for the qualification rate of the human annotations on features drawn from $P_X^{t-1}$. Note that the only difference lies in the third term of the RHS which changes from $\overline{q}_0$ to $q^*_{t-1}$. Our first observation is that $q^*_{t-1}$ is never smaller than $\overline{q}_0$ because the best response will not harm agents' qualifications. With this observation, we can derive Prop. D.1.

**Proposition D.1.** $a'_t \geq a_t$ holds for any $t \geq 1$. If Prop. 3.4 further holds, we also have $a'_t \to 1$.

Prop. D.1 can be proved easily by applying the observation stated above. However, note that unlike $a_t$, $a'_t$ is not necessarily monotonically increasing.

### D.2  Additional discussions on fairness

**Other fairness metrics.** It is difficult to derive concise results considering other fairness metrics including *equal opportunity* and *equal improvability* because the data distributions play a role in determining these metrics. However, Theorem 3.5 states the true qualification rate of the agent population is likely to decrease, suggesting the retraining process may do harm to improvability. Meanwhile, when the acceptance rate $a_t$ increases for the disadvantageous group, the acceptance rate of the qualified individuals will be likely to be better, but it is not guaranteed because the feature distribution of the qualified individuals also changes because we assume the strategic behaviors are causal which may incur label changes.

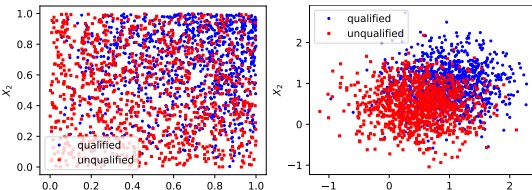

Figure 7: Visualization of distribution: Uniform data (left) and Gaussian data (right)

### D.3 Additional Discussion on the training dataset $\mathcal{S}_t$

$\mathcal{S}_t$ reuses the accumulated data instead of only fine-tuning the model using the available data at the current round. This is more reasonable because: (i) The distribution shifts caused by agent best responses are not known by the decision maker, only using current data may result in forgetting the previous samples that may still be useful. Since agents' best responses will change $Y$ and will not break $P_{Y|X}$, previous samples can provide useful information, especially on the feature domain that current samples do not cover; (ii) the sample size at a single round may be too small even for fine-tuning (e.g., for a college admission example, if we only have tens of people applying during some application round).

Moreover, only using current data does not qualitatively change the theoretical results. Since the theoretical results demonstrate monotonic trends of $a_t, q_t$, only using the most recent data will not alter the trends. We perform an additional set of simulations where the decision maker only uses samples from the most recent round ($S_{m,t-1}, S_{o,t-1}$) on Gaussian Data and Uniform-linear Data with the experimental setups same as the ones in the paper (Section 5 for Gaussian Data and App. E for Uniform-linear Data), and report the average of $a_t, q_t$. Dynamics of $a_t, q_t$ are still similar.

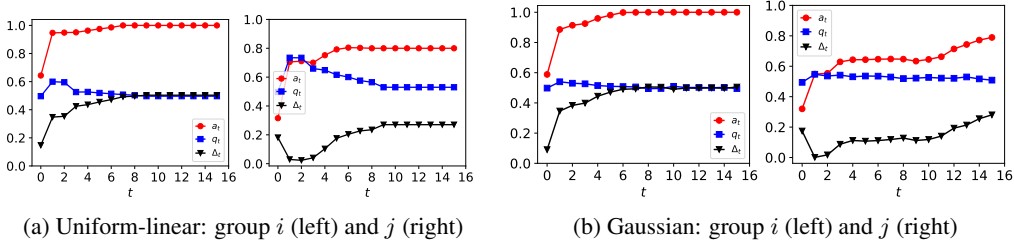

(a) Uniform-linear: group $i$ (left) and $j$ (right)          (b) Gaussian: group $i$ (left) and $j$ (right)

Figure 8: Dynamics of $a_t, q_t, \Delta_t$ when $\mathcal{S}_t$ only contains the most recent samples

### D.4 Additional Discussions on *refined retraining process*

We provide the following proposition to illustrate how the *refined retraining process* leverages a *probabilistic sampler* to stabilize the dynamics.

**Proposition D.2.** *If the decision-maker uses a probabilistic sampler* $h_{t-1}(x) = D_{Y|X}^{t-1}(1|x)$ *to produce model-annotated samples at t, then* $D_{Y|X}^t = D_{Y|X}^o$.

The proof details are in App. G.7. Prop. D.2 illustrates the underlying conditional distribution $D_{Y|X}^t$ is expected to be the same as $D_{Y|X}^o$, meaning that the classifier $f_t$ always learns the distribution of *human-annotated* data, thereby only preserving the systematic bias. However, there is no way to deal with the systematic bias in *refined retraining process*.

### D.5 Additional Discussions on the linearity of the model

Our model is built upon previous works on strategic classification (e.g., [2, 12, 32]) where the decision policy was assumed to be transparent and interpretable since human agents expect to face a linear classifier which they can understand, especially under high-stake situations such as job hiring, college admission and loan application. Meanwhile, as pointed out by Zhang et al. [12], Raab and Liu [13], Levanon and Rosenfeld [67], in practical circumstances, the decision-maker can first fit a non-linear model to learn the embeddings of agents' preliminary features and then produce the embedded features. The decision-maker then uses a linear classifier on the new set of features and

agents also best respond with respect to new features. Under this setting, nonlinearity is involved but our results still hold. Practical examples include (i) FICO credit score [12]: FICO credit score is based on a complex set of features, and the decision-maker (e.g., a bank) simply uses a threshold of the score to assign decisions; (ii) Spam classification [67]: The decision-maker classifies spam using a linear classifier, but the features are produced as an embedding by a neural network based on number of words in the post, number of phone numbers in the post and number of followers of the user.

## E  Main Experiments for other datasets

We provide results on a Uniform dataset and a real dataset [20] with settings similar to Sec. 5.

**Uniform data.** All settings are similar to the Gaussian dataset except that $P_X$ and $P_{Y|X}(1|x)$ change as shown in Table 2.

We first verify the results in Sec. 3 by illustrating the dynamics of $a_t, q_t, \Delta_t$ for both groups (Fig. 9). Since our analysis neglects the algorithmic bias and the evolution results are in expectation, we perform $n = 100$ independent runs of

Table 2: Gaussian Dataset Setting

| $P_{X_k}(x_k)$ | $P_{Y|X}(1|x)$ | $n, r, T, q_0$ |
|---|---|---|
| $\mathcal{U}(0,1)$ | $0.5 \cdot (x_1 + x_2)$ | $100, 0.05, 15, 0.5$ |

experiments for every parameter configuration and show the averaged outcomes. The results are consistent with Thm. 3.3, 3.5 and 3.6: (i) acceptance rate $a_t$ (red curves) increases monotonically; (ii) qualification rate $q_t$ decreases monotonically starting from $t = 1$ (since strategic agents only best respond from $t = 1$); (iii) classifier bias $\Delta_t$ evolves differently for different groups and it may reach the minimum after a few rounds of retraining.

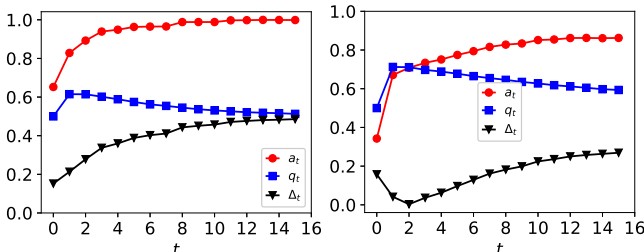

Figure 9: $a_t, q_t, \Delta_t$ for group $i$ (left) and $j$ (right)

Next, we present the results of a set of complementary experiments on real data [20] where we directly fit $P_{X|Y}$ with Beta distributions. The fitting results slightly violate Assumption 3.2. Also $D_X^o$ is not equal to $P_X$. More importantly, logistic models cannot fit these distributions well and produce non-negligible algorithmic bias, thereby violating Assumption 3.1. The following experiments demonstrate how the dynamics change when situations are not ideal.

Table 3: Description of credit approval dataset

| Settings | Group $i$ | Group $j$ |
|---|---|---|
| $P_{X_1|Y}(x_1|1)$ | $Beta(1.37, 3.23)$ | $Beta(1.73, 3.84)$ |
| $P_{X_1|Y}(x_1|0)$ | $Beta(1.50, 4.94)$ | $Beta(1.59, 4.67)$ |
| $P_{X_2|Y}(x_2|1)$ | $Beta(0.83, 2.83)$ | $Beta(0.66, 2.50)$ |
| $P_{X_2|Y}(x_2|0)$ | $Beta(0.84, 5.56)$ | $Beta(0.69, 3.86)$ |
| $n, r, T, q_0$ | $50, 0.05, 15, 0.473$ | $50, 0.05, 15, 10$ |

**Credit approval dataset [20].** We consider credit card applications and adopt the data in UCI Machine Learning Repository processed by Dua and Graff [68]. The dataset includes features of agents from two social groups $i, j$ and their labels indicate whether the credit application is successful. We first preprocess the dataset by normalizing and only keeping a subset of features (two continuous $X_1, X_2$) and labels, then we fit conditional distributions $P_{X_k|Y}$ for each group using Beta distributions (Fig. 10) and calculate prior-best-response qualification rates $q_0^i, q_0^j$ from the dataset. The details are summarized in Table 3. All other parameter settings are the same as the ones of synthetic datasets in Sec. 5.

We first illustrate the dynamics of $a_t, q_t, \Delta_t$ for both groups under different $r$. The results are shown in Fig. 11 and are approximately aligned with Thm. 3.3, 3.5 and 3.6: (i) acceptance rate $a_t$ (red curves) has increasing trends; (ii) qualification rate $q_t$ (blue curves) decreases starting from $t = 1$ (since strategic agents only best respond from $t = 1$); (iii) classifier bias $\Delta_t$ (black curves) evolve differently for different groups.

Next, we illustrate situation (iii) of Thm. 4.1 on this dataset, where the evolutions of unfairness and $\Delta_t$ are shown in Fig. 12. Though the dynamics are still approximately aligned with the theoretical results, the changes are not smooth. However, this is not surprising because several assumptions are violated, and the overall trends still stay the same.

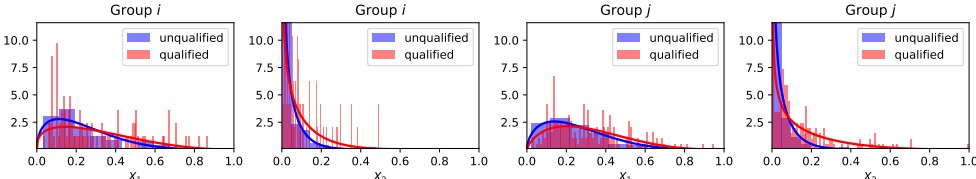

Figure 10: Visualization of distribution for Credit Approval dataset.

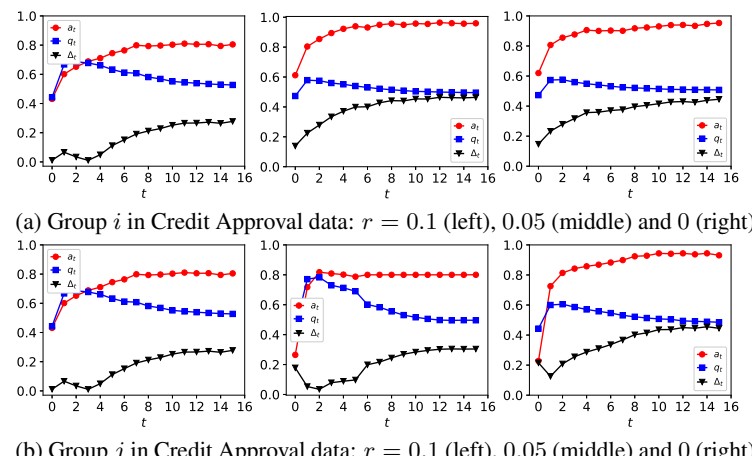

(a) Group $i$ in Credit Approval data: $r = 0.1$ (left), $0.05$ (middle) and $0$ (right)

(b) Group $j$ in Credit Approval data: $r = 0.1$ (left), $0.05$ (middle) and $0$ (right)

Figure 11: Dynamics of $a_t, q_t, \Delta_t$ for Credit Approval dataset

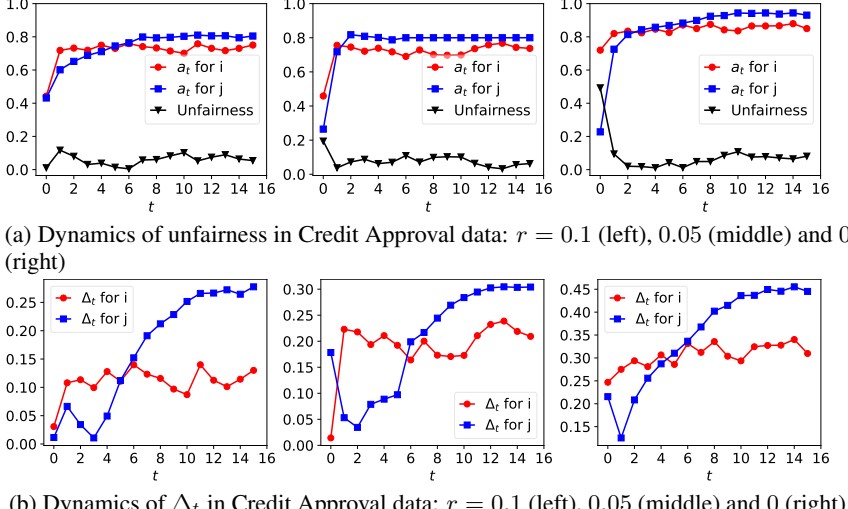

(a) Dynamics of unfairness in Credit Approval data: $r = 0.1$ (left), $0.05$ (middle) and $0$ (right)

(b) Dynamics of $\Delta_t$ in Credit Approval data: $r = 0.1$ (left), $0.05$ (middle) and $0$ (right)

Figure 12: Dynamics of unfairness and $\Delta_t$ for Credit Approval dataset

# F  Additional Results on Synthetic/Semi-synthetic Datasets

In this section, we provide comprehensive experimental results conducted on two synthetic datasets and one semi-synthetic dataset mentioned in Sec. 5 of the main paper. Specifically, App. F.1 gives

the details of experimental setups; App. F.2 demonstrates additional results to verify Theorem 3.3 to Theorem 3.6 under different $r$ (i.e., ratios of human-annotated examples available at each round). The section also gives results on how $a_t, q_t, \Delta_t$ change under a long time horizon or when there is no systematic bias; App. F.3 further demonstrates the results under *refined retraining process*; App. F.4 provides fairness dynamics under various values of $r$ on different datasets; App. F.5 illustrates how the results in the main paper still hold when strategic agents have noisy best responses; App. F.6 compares the situations when agents are non-strategic with the ones when they are strategic, demonstrating how agents' strategic behaviors produce more extreme dynamics of $a_t, q_t, \Delta_t$.

## F.1 Additional Experimental Setups

Generally, we run all experiments on a MacBook Pro with Apple M1 Pro chips, memory of 16GB and Python 3.9.13. All experiments are randomized with seed 42 to run $n$ rounds. Error bars are provided in App. H. All experiments train $f_t$ with a logistic classifier using SGD as its optimizer. Specifically, we use `SGDClassifier` with `logloss` to fit models.

**Synthetic datasets.** The basic description of synthetic datasets 1 and 2 is shown in Sec. 5 and App. E. We further provide the visualizations of their distributions in Fig. 7.

**German Credit dataset.** There are 2 versions of the German Credit dataset according to UCI Machine Learning Database [19], and we are using the one where all features are numeric. Firstly, we produce the sensitive features by ignoring the marital status while only focusing on sex. Secondly, we use `MinMaxScaler` to normalize all features. The logistic model itself can satisfy Assumption 3.2 with minimal operations: if feature $i$ has coefficients smaller than 0, then just negate it and the coefficients will be larger than 0 and satisfy the assumption.

## F.2 Additional Results to Verify Thm. 3.3 to Thm. 3.6

**Dynamics under different $r$.** Although experiments in Sec. 5 and App. E already demonstrate the validity of Thm. 3.3, Prop. 3.4 and Thm. 3.6, the ratio $r$ of human-annotated examples is subject to change in reality. Therefore, we first provide results for $r \in \{0, 0.05, 0.1, 0.3\}$ in all 3 datasets. $r$ only has small values since human-annotated examples are likely to be expensive to acquire. Fig. 17 shows all results under different $r$ values. Specifically, Fig. 17a and 17b show results for synthetic dataset 1, Fig. 17c and 17d show results for synthetic dataset 2, while Fig. 17e and 17f show results for German Credit data. On every row, $r = 0.3, 0.1, 0.05, 0$ from the left to the right. All figures demonstrate the robustness of the theoretical results, where $a_t$ always increases and $q_t$ decreases starting from $t = 1$. $\Delta_t$ also has different dynamics as specified in Thm. 3.6.

**Dynamics under a long time horizon to verify Prop. 3.4.** Prop. 3.4 demonstrates that when $r$ is small enough, $a_t$ will increase towards 1. Therefore, we provide results in all 3 datasets when $r = 0$ to see whether $a_t$ increases to be close to 1. As Fig. 15 shows, $a_t$ is close to 1 after tens of rounds, validating Prop. 3.4.

**Dynamics under a different $B$.** Moreover, individuals may incur different costs to alter different features, so we also provide the dynamics of $a_t, q_t, \Delta_t$ when the cost matrix $B = \begin{bmatrix} 3 & 0 \\ 0 & 6 \end{bmatrix}$ in two synthetic datasets. Fig. 16 shows the differences in costs of changing different features do not affect the theoretical results.

**Dynamics when all samples are human-annotated.** Though this is unlikely to happen under the Strategic Classification setting as justified in the main paper, we provide an illustration when all training examples are *human-annotated* (i.e., $r = 1$) when humans systematically overestimate the qualification in both synthetic datasets. Theoretically, the difference between $a_t$ and $q_t$ should be relatively consistent, which means $\Delta_t$ is only due to the systematic bias. Fig. 15d verifies this.

## F.3 Additional Results on *refined retraining process*

In this section, we provide more experimental results demonstrating how *refined retraining process* stabilizes the dynamics of $a_t, q_t, \Delta_t$ but still preserves the systematic bias. Specifically, we produce plots similar to Fig. 17 in Fig. 18, but the only difference is that we use probabilistic samplers for

model-annotated examples. From Fig. 18, it is obvious the deviations of $a_t$ from $q_t$ have the same directions and approximately the same magnitudes as the systematic bias.

## F.4 Additional Results on Fairness

In this section, we provide additional results on the dynamics of unfairness and classifier bias under different $r$ (the same settings in App. F.2). We aim to illustrate situation (iii) in Thm. 4.1 where the *refined retraining process* is applied on the advantaged group $i$ while the original process is applied on the disadvantaged group $j$. From Fig. 19a, 19c and 19e, we can see unfairness reaches a minimum in the middle of the retraining process, suggesting the earlier stopping of retraining brings benefits. From Fig. 19b, 19d and 19f, we can see $\Delta_t$ for the disadvantaged group $j$ reaches a minimum in the middle of the retraining process, but generally not at the same time when unfairness reaches a minimum.

## F.5 Additional Results on Noisy Best Responses

Following the discussion in Sec. 5, we provide dynamics of $a_t, q_t, \Delta_t$ of both groups under different $r$ similar to App. F.2 but when the agents have noisy knowledge. The only difference is that the agents' best responses are noisy in that they only know a noisy version of classification outcomes: $\widetilde{f}_t(x) = f_t(x) + \epsilon$, where $\epsilon$ is a Gaussian noise with mean $0$ and standard deviation $0.1$. Fig.20 shows that Thm. 3.3 to Thm. 3.6 are still valid.

## F.6 Comparisons between Strategic and Non-strategic Situations

In this section, we show the absence of strategic behaviors may result in much more consistent dynamics of $a_t, q_t, \Delta_t$ as illustrated in Fig. 21.

## F.7 Additional Results on a Dataset with Non-Linearity.

ACSIncome-CA dataset [69] is a larger dataset consisting of over $150K$ records for agents and their annual income. The decision-maker wants to predict whether a person has an annual income $> 50000$. We assume the decision-maker first trains a 2-d embedding using a neural network and 53 original features then regards the embedding as the new feature. We divide the agents into 2 groups based on their ages. Similar to the credit approval dataset, we then fit Beta distributions on the 2 groups and then verify the monotonic likelihood assumption (Fig. 13). We then plot the dynamics of $a_t, q_t, \Delta_t$ for both groups when the systematic bias is either positive or negative. The results show that similar trends still hold for this large dataset (Fig. 14).

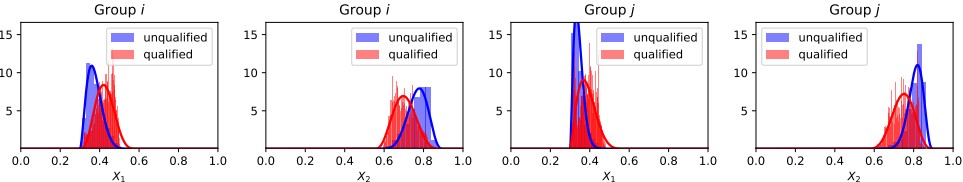

Figure 13: Feature distribution of Income Dataset.

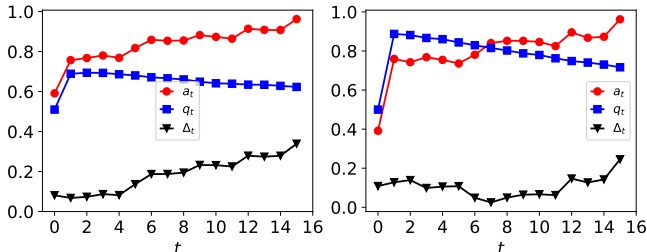

Figure 14: Dynamics of $a_t, q_t, \Delta_t$ of the income dataset: the left plot is for group $i$ and the right plot is for group $j$. We run 10 trials for each experiment and agent cost matrices are the same as the experiments in the main paper. The results are similar to the ones in the main paper.

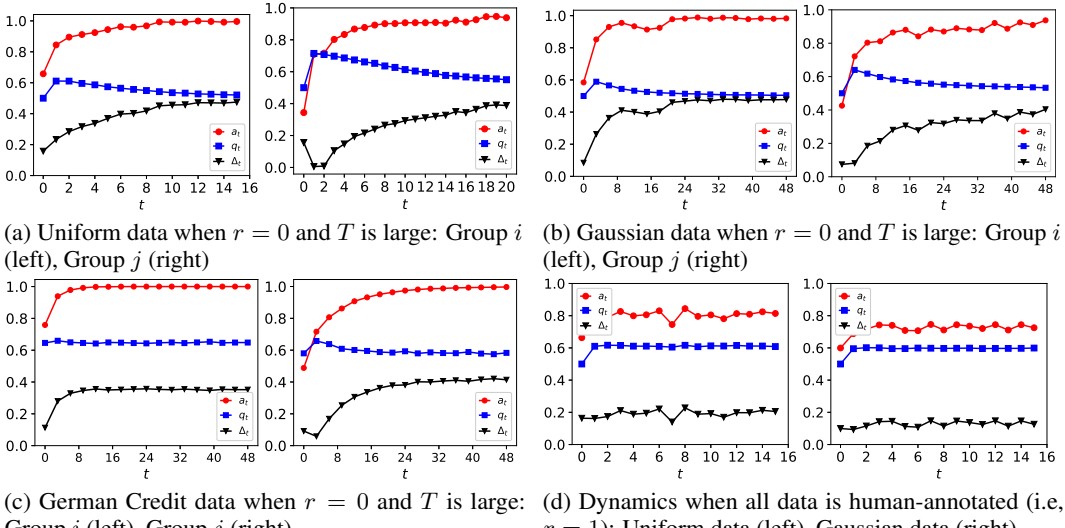

(a) Uniform data when $r = 0$ and $T$ is large: Group $i$ (left), Group $j$ (right)

(b) Gaussian data when $r = 0$ and $T$ is large: Group $i$ (left), Group $j$ (right)

(c) German Credit data when $r = 0$ and $T$ is large: Group $i$ (left), Group $j$ (right)

(d) Dynamics when all data is human-annotated (i.e, $r = 1$): Uniform data (left), Gaussian data (right).

Figure 15: Dynamics of $a_t, q_t, \Delta_t$ on all datasets when $r = 0$ and $T$ is large or when all examples are annotated by humans.

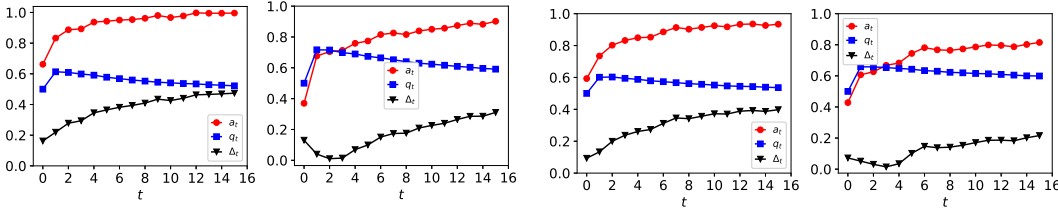

(a) Uniform data with a different $B$ : Group $i$ (left), Group $j$ (right)

(b) Gaussian data with a different $B$ : Group $i$ (left), Group $j$ (right)

Figure 16: Dynamics of $a_t, q_t, \Delta_t$ on synthetic datasets. Except $B$, all other settings are as same as in Sec. 5.

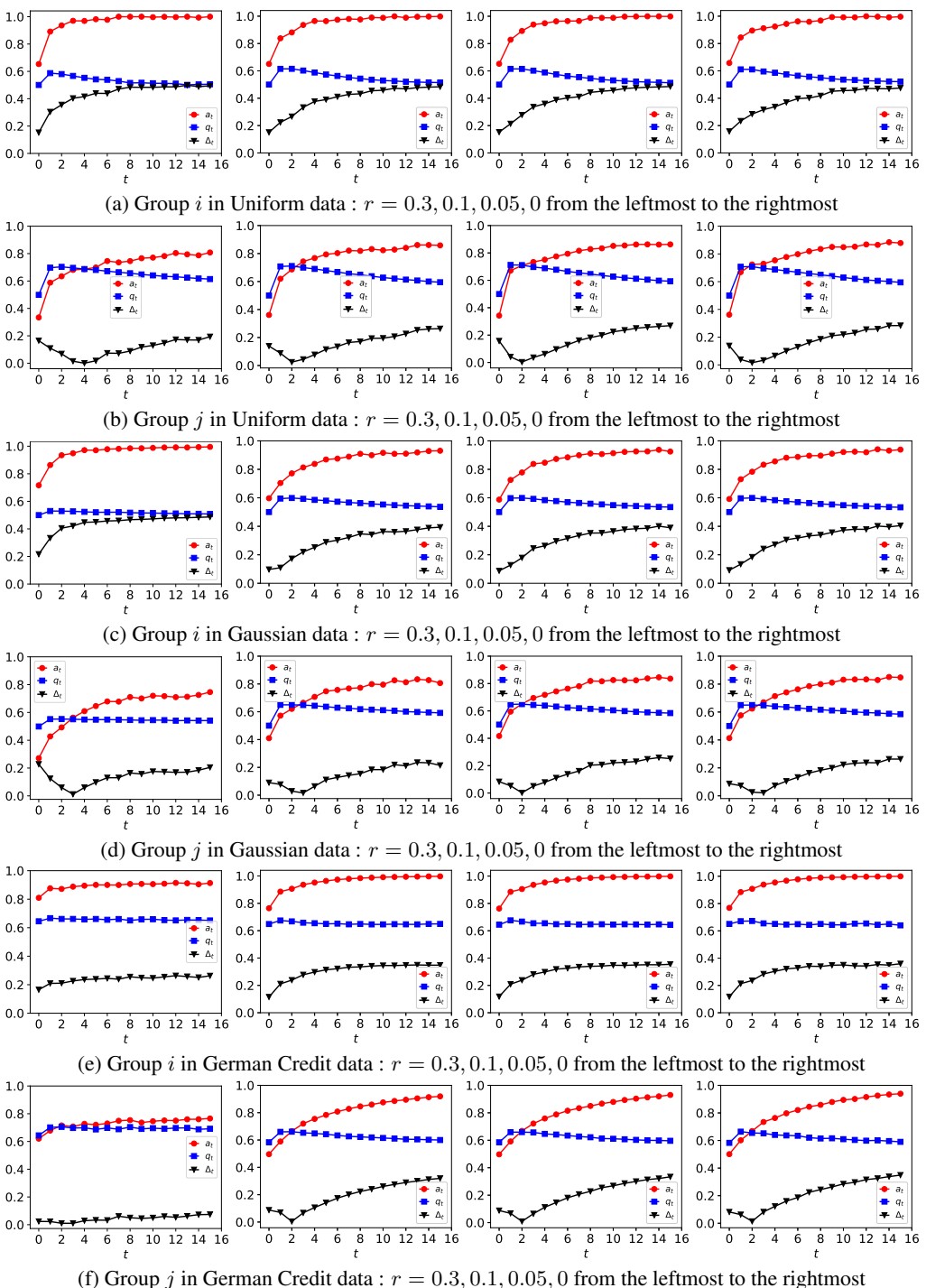

(a) Group $i$ in Uniform data : $r = 0.3, 0.1, 0.05, 0$ from the leftmost to the rightmost

(b) Group $j$ in Uniform data : $r = 0.3, 0.1, 0.05, 0$ from the leftmost to the rightmost

(c) Group $i$ in Gaussian data : $r = 0.3, 0.1, 0.05, 0$ from the leftmost to the rightmost

(d) Group $j$ in Gaussian data : $r = 0.3, 0.1, 0.05, 0$ from the leftmost to the rightmost

(e) Group $i$ in German Credit data : $r = 0.3, 0.1, 0.05, 0$ from the leftmost to the rightmost

(f) Group $j$ in German Credit data : $r = 0.3, 0.1, 0.05, 0$ from the leftmost to the rightmost

Figure 17: Dynamics of $a_t, q_t, \Delta_t$ on all datasets.

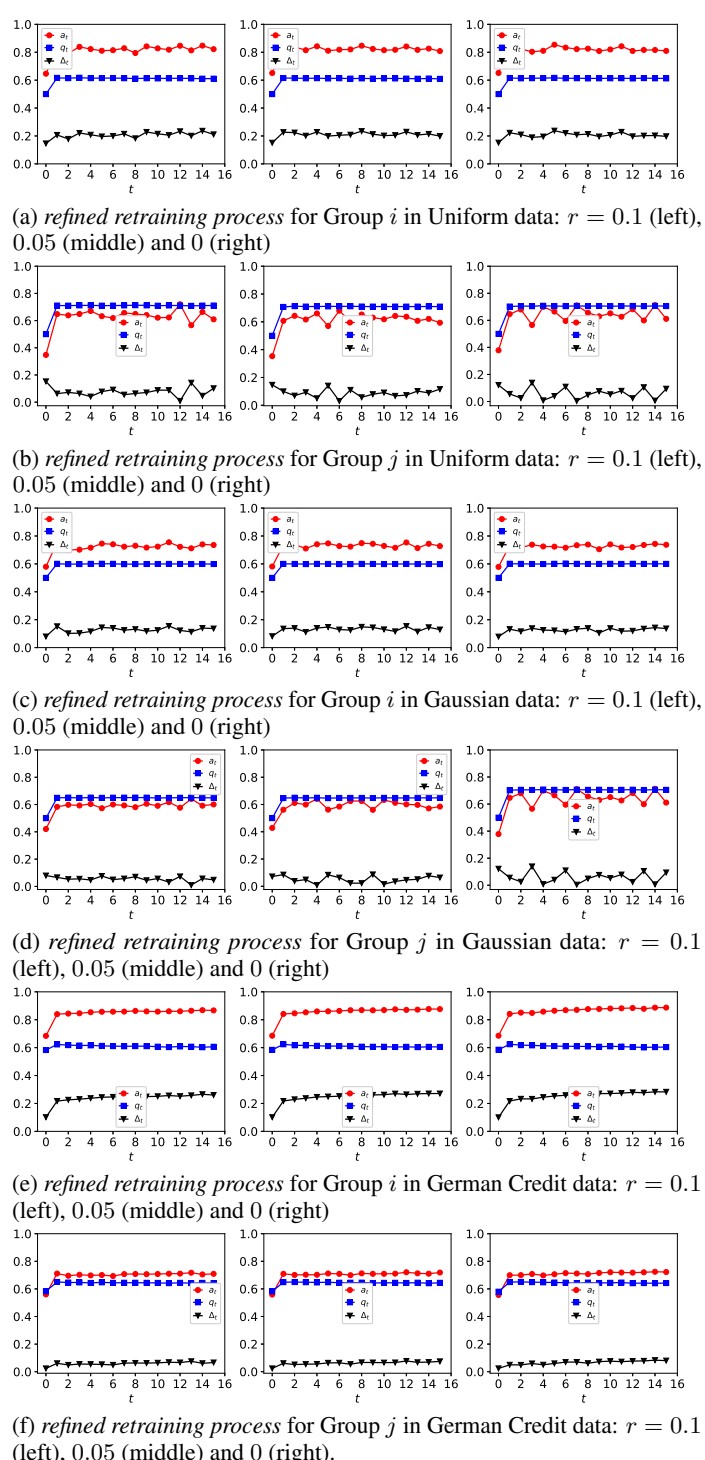

(a) *refined retraining process* for Group $i$ in Uniform data: $r = 0.1$ (left), 0.05 (middle) and 0 (right)

(b) *refined retraining process* for Group $j$ in Uniform data: $r = 0.1$ (left), 0.05 (middle) and 0 (right)

(c) *refined retraining process* for Group $i$ in Gaussian data: $r = 0.1$ (left), 0.05 (middle) and 0 (right)

(d) *refined retraining process* for Group $j$ in Gaussian data: $r = 0.1$ (left), 0.05 (middle) and 0 (right)

(e) *refined retraining process* for Group $i$ in German Credit data: $r = 0.1$ (left), 0.05 (middle) and 0 (right)

(f) *refined retraining process* for Group $j$ in German Credit data: $r = 0.1$ (left), 0.05 (middle) and 0 (right).

Figure 18: Illustrations of *refined retraining process* on all 3 datasets.

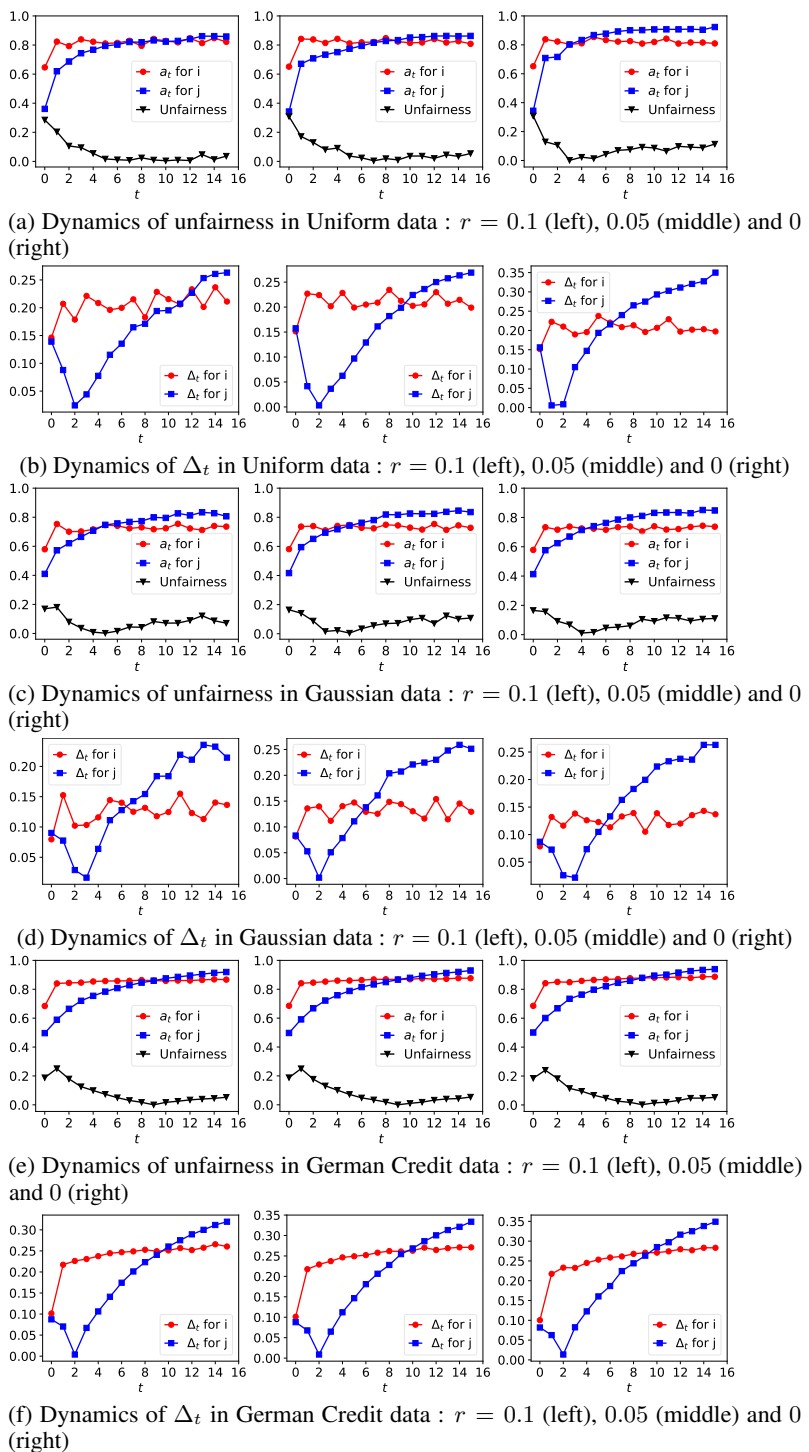

(a) Dynamics of unfairness in Uniform data : $r = 0.1$ (left), 0.05 (middle) and 0 (right)

(b) Dynamics of $\Delta_t$ in Uniform data : $r = 0.1$ (left), 0.05 (middle) and 0 (right)

(c) Dynamics of unfairness in Gaussian data : $r = 0.1$ (left), 0.05 (middle) and 0 (right)

(d) Dynamics of $\Delta_t$ in Gaussian data : $r = 0.1$ (left), 0.05 (middle) and 0 (right)

(e) Dynamics of unfairness in German Credit data : $r = 0.1$ (left), 0.05 (middle) and 0 (right)

(f) Dynamics of $\Delta_t$ in German Credit data : $r = 0.1$ (left), 0.05 (middle) and 0 (right)

Figure 19: Dynamics of unfairness and $\Delta_t$ of all datasets.

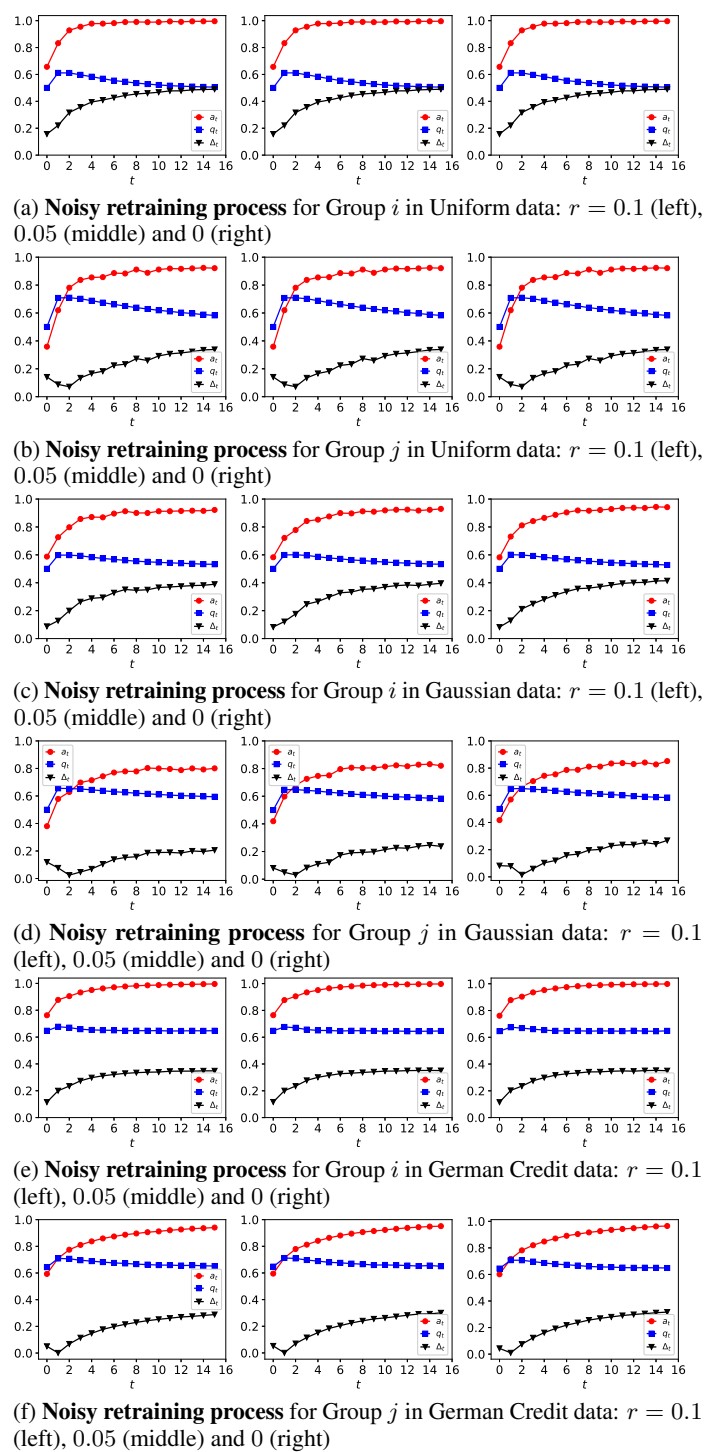

(a) **Noisy retraining process** for Group $i$ in Uniform data: $r = 0.1$ (left), 0.05 (middle) and 0 (right)

(b) **Noisy retraining process** for Group $j$ in Uniform data: $r = 0.1$ (left), 0.05 (middle) and 0 (right)

(c) **Noisy retraining process** for Group $i$ in Gaussian data: $r = 0.1$ (left), 0.05 (middle) and 0 (right)

(d) **Noisy retraining process** for Group $j$ in Gaussian data: $r = 0.1$ (left), 0.05 (middle) and 0 (right)

(e) **Noisy retraining process** for Group $i$ in German Credit data: $r = 0.1$ (left), 0.05 (middle) and 0 (right)

(f) **Noisy retraining process** for Group $j$ in German Credit data: $r = 0.1$ (left), 0.05 (middle) and 0 (right)

Figure 20: Illustrations of **noisy retraining process** on all 3 datasets.

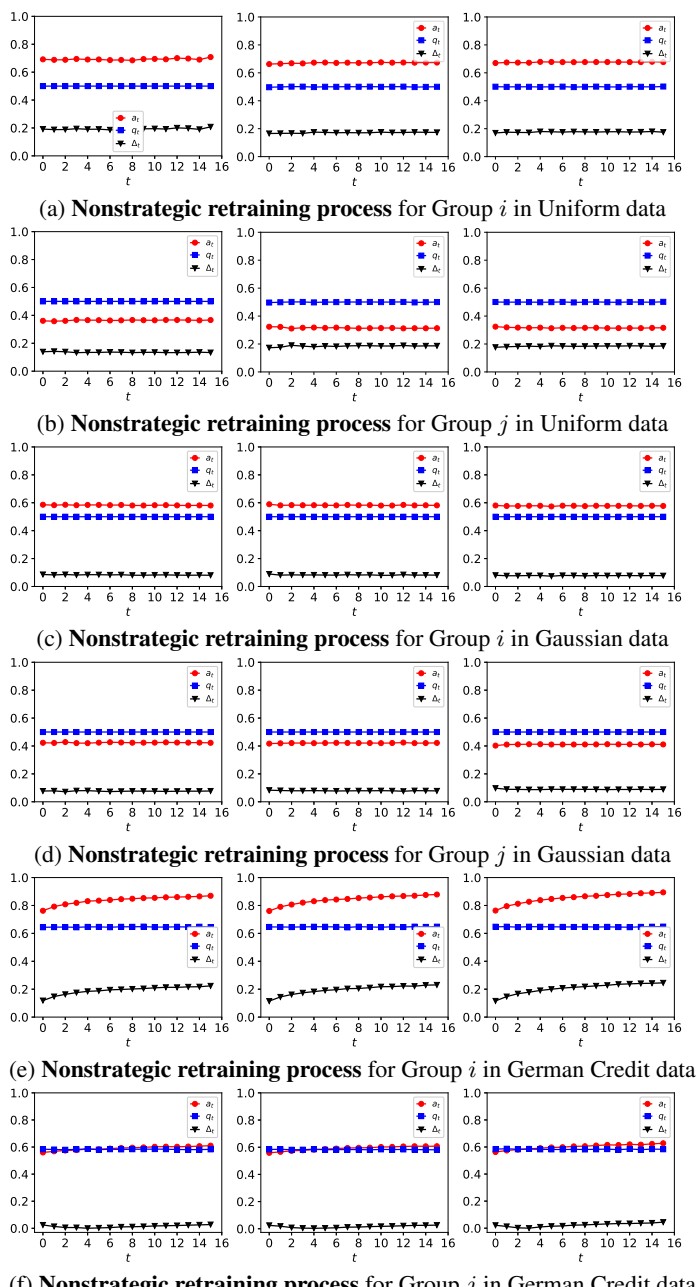

(a) **Nonstrategic retraining process** for Group $i$ in Uniform data

(b) **Nonstrategic retraining process** for Group $j$ in Uniform data

(c) **Nonstrategic retraining process** for Group $i$ in Gaussian data

(d) **Nonstrategic retraining process** for Group $j$ in Gaussian data

(e) **Nonstrategic retraining process** for Group $i$ in German Credit data

(f) **Nonstrategic retraining process** for Group $j$ in German Credit data

Figure 21: Illustrations of **nonstrategic retraining process** on all 3 datasets: $r = 0.1$ (left), 0.05 (middle) and 0 (right)

# G   Derivations and Proofs

## G.1   Dynamics of the Expected Qualification Rate of $\mathcal{S}_t$

Let us begin by defining $\overline{q}_t := \mathbb{E}_{\mathcal{S}_t}[Q(\mathcal{S}_t)]$ which is the expected qualification rate of the training dataset at $t$. While it is difficult to derive the dynamics of $a_t$ and $q_t$ explicitly, we can first work out the dynamics of $\overline{q}_t$ using the law of total probability (details in App. G.1), i.e.,

$$\overline{q}_t = \tfrac{tN+(t-1)K}{(t+1)N+tK} \cdot \overline{q}_{t-1} + \tfrac{N}{(t+1)N+tK} \cdot a_{t-1} + \tfrac{K}{(t+1)N+tK} \cdot \overline{q}_0 \tag{4}$$

To derive $\overline{q}_t := \mathbb{E}_{\mathcal{S}_t}[Q(\mathcal{S}_t)]$, we first refer to (2) to get $|\mathcal{S}_t| = (t+1)N + tK$. Then, by the law of total probability, the expected qualification rate of $\mathcal{S}_t$ equals to the weighted sum of the expected qualification rate of $\mathcal{S}_{t-1}$, $S_{o,t-1}$ and the expectation of $f_{t-1}(x)$ over $\mathcal{S}_{m,t-1}$ as follows:

$$\tfrac{tN+(t-1)K}{(t+1)N+tK} \mathbb{E}_{\mathcal{S}_{t-1}}[Q(\mathcal{S}_{t-1})] + \tfrac{N}{(t+1)N+tK} + \mathbb{E}_{\mathcal{S}_{t-1}}[A(f_{t-1}, P^{t-1})] + \tfrac{K}{(t+1)N+tK} \mathbb{E}_{\mathcal{S}_{o,t-1}}[Q(\mathcal{S}_{o,t-1})]$$

The second expectation is exactly the definition of $a_{t-1}$. Moreover, note that $Q(\mathcal{S}_{o,t-1}) = Q(\mathcal{S}_{o,0}) = Q(\mathcal{S}_0) = \overline{q}_0$ for any $t > 0$, the third expectation is exactly $\overline{q}_0$, so the above equation is exactly (4).

## G.2   Derivation of factors influencing $a_t, q_t$

As stated in Sec.2, we can get the factors influencing the evolution of $a_t$, $q_t$ by finding all sources affecting $P_{XY}^t$ and the expectation of $f_t(x)$ over $P_X^t$.

We first work out the sources influencing $f_t$ and the expectation of $f_t(x)$ over $P^t(X)$:

- $\overline{q}_t$: since $f_t$ is trained with $\mathcal{S}_t$, $\overline{q}_t$ is a key factor influencing the classifier.
- $h_t$ may not model $D_{Y|X}^t$ accurately enough. However, we ignore this by Assumption 3.1, where realizability is a common assumption in theoretical proof, while the experiments do not ignore this source.
- $\delta_{BR}^t$: we now know factors influencing the expectation of $f_t(x)$ over $D_X^t$, then the only left factor is the ones accounting for the difference between $D_X^t$ and $P_X^t$. Note that only the best responses of agents can change the marginal distribution $P_X$. We then denote it as $\delta_{BR}^t$.

Then, with $P_{XY}$ known, $P_{XY}^t$ is only influenced by $f_{t-1}$ (i.e., the agents' best responses to the classifier at $t-1$). $f_{t-1}$ is also dependent on the above factors. So we get all factors.

## G.3   Proof of Theorem 3.3

proving the following lemma:

**Lemma G.1.** *Assume $t \geq 2$ and the following conditions hold: (i) $\overline{q}_t > \overline{q}_{t-1} \geq \overline{q}_{t-2}$; (ii) $\forall x \in X$, $D_{Y|X}^t(1|x) \geq D_{Y|X}^{t-1}(1|x)$; (iii) $\forall x, \overline{f}_{t-1}(x) \geq \overline{f}_{t-2}(x)$. Let $\overline{f}_{t-1} = \mathbb{E}_{\mathcal{S}_{t-1} \sim D_{XY}^{t-1}}[f_{t-1}], \overline{f}_t = \mathbb{E}_{\mathcal{S}_t \sim D_{XY}^t}[f_t]$, we have the following results:*

*1. $\forall x, \overline{f}_t(x) \geq \overline{f}_{t-1}(x)$.*

*2. There exists a non-zero measure subset of $x$ values that satisfies the strict inequality.*

**Proof.** We first prove (i). Note that we assume $h_t$ models $D_{Y|X}^t$ well in Assumption 3.1, so $\overline{f}_{t-1}$ (resp. $\overline{f}_t$) outputs 1 if $D_{Y|X}^{t-1}(1|x)$ (resp. $D_{Y|X}^t(1|x)$) is larger than some threshold $\theta$. Then, according to the above condition (ii), $\forall \theta$, if $D_{Y|X}^{t-1}(1|x) > \theta$, $D_{Y|X}^t(1|x) \geq D_{Y|X}^{t-1}(1|x) > \theta$. This demonstrates that $\overline{f}_{t-1}(x) = 1$ implies $\overline{f}_t(x) = 1$ and (i) is proved.

Next, according to (2), if $\overline{q}_t > \overline{q}_{t-1}$, this means $\mathbb{E}_{y \sim D_Y^t}[y] > \mathbb{E}_{y \sim D_Y^t}[y]$. Since $D_Y^t = D_X^t \cdot D_{Y|X}^t$, **either** there exists at least one non-zero measure subset of $x$ values satisfying $D_{Y|X}^t(1|x) > D_{Y|X}^{t-1}(1|x)$ **or** $D_X^t$ is more "skewed" to the larger values of $x$ (because of monotonic likelihood assumption 3.2). For the second possibility, note that the only possible cause for the feature distribution in

the training dataset $D_X^t$ to gain such a skewness is agents' strategic behaviors. However, since $\overline{f}_{t-1}(x) \geq \overline{f}_{t-2}(x)$ always holds, $\overline{f}_{t-1}$ sets a lower admission standard where some $x$ values that are able to best respond to $\overline{f}_{t-2}$ and improve will not best respond to $\overline{f}_{t-1}$, thereby impossible to result in a feature distribution shift to larger $x$ values while keeping the conditional distribution unchanged. Thus, only the first possibility holds. $\qquad\square$

Then we prove Theorem 3.3 using mathematical induction to prove a stronger version:

**Lemma G.2.** *When $t > 1$, $\overline{q}_t > \overline{q}_{t-1}$, $D_{Y|X}^t(1|x) \geq D_{Y|X}^{t-1}(1|x)$, $\overline{f}_t(x) \geq \overline{f}_{t-1}(x)$, and finally $a_t > a_{t-1}$.*

**Proof.** $t$ starts from 2, but we need to prove the following claim: $\overline{q}_1$ is "almost equal" to $\overline{q}_0$, so are $D_{Y|X}^1$ and $D_{Y|X}^0$.

Firstly, according to the law of total probability, we can derive $\overline{q}_1$ as follows:

$$\overline{q}_1 = \frac{N}{2N+K} \cdot \overline{q}_0 + \frac{N}{2N+K} \cdot a_0 + \frac{K}{2N+K} \cdot \overline{q}_0 \tag{5}$$

The first and the third element are already multiples of $\overline{q}_0$. Also, we know $D_X^o = P_X$. Then, since $D_{Y|X}^t(1|x)$ falls in $\mathcal{H}$ and agents at $t = 0$ have no chance to best respond, we have $a_0 = \mathbb{E}_{X \sim P_X}[D_{Y|X}^o(1|x)] = \overline{q}_0$. Thus, $a_0$ is also equal to $\overline{q}_0$, and the claim is proved. Still, as Assumption 3.1 assumes realizability of $h_t$, $\overline{f}_1$ is the same as $\overline{f}_0$.

**Next, we can prove the lemma by induction**:

1. Base case: Similar to Eq.5, we are able to derive $\overline{q}_2$ as follows:

$$\overline{q}_2 = \frac{2N+K}{3N+2K} \cdot \overline{q}_1 + \frac{N}{3N+2K} \cdot a_1 + \frac{K}{3N+2K} \cdot \overline{q}_0 \tag{6}$$

Based on the claim above, we can just regard $\overline{q}_0$ as $\overline{q}_1$. Then we may only focus on the second term. Since $\overline{q}_1$ and $\overline{q}_0$ are "almost equal" and both distributions should satisfy the monotonic likelihood assumption 3.2, we can conclude $\overline{f}_0$, $\overline{f}_1$ are "almost identical". Then the best responses of agents to $\overline{f}_0$ will also make them be classified as 1 by $\overline{f}_1$, and this will directly ensure the second term to be $A(f_1, P^1) > A(f^1, P) = A(f^0, P)$. The "larger than" relationship is because strategic best responses at the first round will only enable more agents to be admitted. Thus, the first and the third term stay the same as $\overline{q}_1$ while the second is larger, so we can claim $\overline{q}_2 > \overline{q}_1$. Moreover, the difference between $D_{XY}^2$ and $D_{XY}^1$ are purely produced by the best responses at $t = 1$, which will never decrease the conditional probability of $y = 1$. Thus, $D_{Y|X}^2(1|x) \geq D_{Y|X}^1(1|x)$. Together with $\overline{f}_1 = \overline{f}_0$, all three conditions in Lemma G.1 are satisfied. we thereby claim that for every $x$ admitted by $\overline{f}_1$, $\overline{f}_2(x) \geq \overline{f}_1(x)$ and there exists some $x$ satisfying the strict inequality. Note that $P^1$ is expected to be the same as $P^2$ since $f_1 = f_0$. Thus, $A(f_2, P^2) > A(f_1, P^2) = A(f_1, P^1)$, which is $a_2 > a_1$. The base case is proved.

2. Induction step: To simplify the notion, we can write:

$$\begin{aligned} \overline{q}_t &= \frac{tN+(t-1)K}{(t+1)N+tK} \cdot \overline{q}_{t-1} + \frac{N}{(t+1)N+tK} \cdot a_{t-1} + \frac{K}{(t+1)N+tK} \cdot \overline{q}_0 \\ &= A_t \cdot \overline{q}_{t-1} + B_t \cdot a_{t-1} + C_t \cdot \overline{q}_0 \end{aligned}$$

Note that $\overline{q}_{t-1}$ can also be decomposed into three terms:

$$\begin{aligned} \overline{q}_{t-1} &= \frac{(t-1)N+(t-2)K}{tN+(t-1)K} \cdot \overline{q}_{t-2} + \frac{N}{tN+(t-1)K} \cdot a_{t-2} + \frac{K}{tN+(t-1)K} \cdot \overline{q}_0 \\ &= A_{t-1} \cdot \overline{q}_{t-1} + B_{t-1} \cdot a_{t-2} + C_{t-1} \cdot \overline{q}_0 \end{aligned}$$

Since the expectation in the second term is just $a_{t-1}$, and we already know $a_{t-1} > a_{t-2}$, we know $B_t \cdot a_{t-1} + C_t \cdot \overline{q}_0 > B_t \cdot a_{t-2} + C_t \cdot \overline{q}_0$. Note that $\frac{B_t}{B_{t-1}} = \frac{C_t}{C_{t-1}}$, we let the ratio be $m < 1$. Then since $B_{t-1} \cdot a_{t-2} + C_{t-1} \cdot \overline{q}_0 > (B_{t-1} + C_{t-1}) \cdot \overline{q}_{t-1}$ due to $\overline{q}_{t-1} > \overline{q}_{t-2}$, we can derive $B_t \cdot a_{t-1} + C_t \cdot \overline{q}_0 > m \cdot (B_{t-1} + C_{t-1}) \cdot \overline{q}_{t-1} = (B_t + C_t) \cdot \overline{q}_{t-1}$. Then $\overline{q}_t > (A_t + B_t + C_t) \cdot \overline{q}_{t-1} = \overline{q}_{t-1}$. The first claim is proved. As $a_{t-1} > a_{t-2}$ and $D^o$ stays the same, any agent will not have a less probability of being qualified in $\mathcal{S}_t$ compared to in $\mathcal{S}_{t-1}$, demonstrating $D^t_{Y|X}(1|x) \geq D^{t-1}_{Y|X}(1|x)$ still holds. And similarly, we can apply Lemma G.1 to get $\overline{f}_t(x) \geq \overline{f}_{t-1}(x)$ and $a_t > a_{t-1}$. $\qquad\square$

Now we already prove Lemma G.2, which already includes Theorem 3.3. We also want to note here, the proof of Theorem 3.3 does not rely on the initial $\overline{q}_0$ which means it holds regardless of the systematic bias.

### G.4 Proof of Proposition 3.4

We prove the proposition by considering two extreme cases:

(i) When $\frac{K}{N} \to 0$, this means we have no decision-maker annotated sample coming in each round and all new samples come from the deployed model. We prove $\lim_{t\to\infty} a_t = 1$ by contradiction: firstly, by *Monotone Convergence Theorem*, the limit must exist because $a_t > a_{t-1}$ and $a_t < 1$. Let us assume the limit is $\overline{a} < 1$. Then, since $K = 0$, when $t \to \infty$, $\overline{q}_t$ will also approach $\overline{a}$, this means the strategic shift $\delta^t_{BR}$ approaches 0. However, this shift only approaches 0 when all agents are accepted by $f_{t-1}$ because otherwise there will be a proportion of agents benefiting from best responding to $f_{t-1}$ and result in a larger $\overline{q}_{t+1}$ Thus, the classifier at $t+1$ will admit more people and the stability is broken. This means the only possibility is $lim_{t\to\infty} a_{t-1} = 1$ and produces a conflict.

(ii) When $\frac{K}{N} \to \infty$, the problem shrinks to retrain the classifier to fit $D^o_{XY}$, this will make $a_t = a_0$.

Thus, there exists some threshold $\lambda$, when $\frac{K}{N} < \lambda$, $lim_{t\to\infty} a_t = 1$. In practice, the $\lambda$ could be very small. $\qquad\square$

### G.5 Proof of Theorem 3.5

Firstly, since $P^t_{XY}$ differs from $P_{XY}$ only because agents' best respond to $f_t$, we can write $q_t = q_0 + r_t$ where $r_t$ is the difference of qualification rate caused by agents' best responses to $f_t$. Qualitatively, $r_t$ is completely determined by two sources: (i) the proportion of agents who move their features when they best respond; (ii) the increase in the probability of being qualified for each agent. Specifically, each agent that moves its features increases its probability of being qualified from its initial point to a point at the decision boundary of $f_t$. For an agent with initial feature $x$ and improved feature $x^*$, its improvement can be expressed as $U(x) = P_{Y|X}(1|x^*) - P_{Y|X}(1|x)$.

Next, denote the Euclidean distance between $x$ and the decision boundary of $f_t$ as $d_{x,t}$. Noticing that the agents will best respond to the decision classifier if and only if the Euclidean distance between her feature vector and the decision boundary is less than or equal to some constant $C$ no matter where the boundary is (Lemma 2 in [2]) and what the cost matrix $B$ is, we can express the total improvement at $t$ as follows:

$$I(t) = \int_{d_{x,t} \leq C} P_X(x) \cdot U(x) \, dx \tag{7}$$

According to the proof of Theorem 3.3, all agents with feature vector $x$ who are admitted by $f_0$ will be admitted by $f_t$ ($t > 0$), making all agents who possibly improve must belong to $\mathcal{J}$. When $F_X$ is convex in $\mathcal{J}$ or it is convex in each dimension separately, we know $P_X$ as its derivative will be non-decreasing in each of the $d$ dimensions within $\mathcal{J}$. Similarly, when $P_{Y|X}(1|x)$ is convex in each of its dimensions or the whole $\mathcal{J}$, $U$ is also non-decreasing in each of the $d$ dimensions within $\mathcal{J}$. Then note that $f_t$ always lies below $f_{t-1}$, therefore, for each agent who improves at $t$ having feature vector $x_{i,t}$, we can find an agent at $t-1$ with corresponding $x_{i,t-1}$ such that both $P_X(x_{i,t-1}) \geq P_X(x_{i,t})$ and $U(x_{i,t-1}) \geq U(x_{i,t})$. This will ensure $I(t) \leq I(t-1)$. Thus, $q_t$ decreases starting from 1. $\qquad\square$

## G.6 Proof of Theorem 3.6

- $\mu(D^o, P) \geq 0$: according to Def. 2.1, $a_0 = \mathbb{E}_{P_X}[D^o_{Y|X}(1|x)] > \mathbb{E}_{P_X}[P_{Y|X}(1|x)] = Q(P) = q_0$. Now that $a_0 - q_0 \geq 0$. Based on Thm. 3.3 and Thm. 3.5, $a_t$ is increasing, while $q_0$ is decreasing, so $\Delta_t$ is always increasing.

- $\mu(D^o, P) < 0$: similarly we can derive $a_0 - q_0 < 0$, so $\Delta_0 = |a_0 - q_0| = q_0 - a_0$. So while $a_0 - q_0$ is still increasing, $\Delta_t$ will first decrease. Moreover, according to Prop. 3.4, if $\frac{K}{N}$ is small enough, $\Delta_t$ will eventually exceed 0 and become larger again. Thus, $\Delta_t$ either decreases or first decreases and then increases. $\qquad\square$

## G.7 Proof of Proposition D.2

Firstly, $\mathcal{S}_0 = \mathcal{S}_{o,0}$ and $\mathcal{S}_1 = \mathcal{S}_0 \cup \mathcal{S}_{o,1} \cup \mathcal{S}_{m,0}$. Obviously, the first two sets are drawn from $D^o_{XY}$. Consider $\mathcal{S}_{m,0}$, since it is now produced by labeling features from $P_X = D^o_X$ with $D^o_{Y|X}$, $\mathcal{S}_{m,0}$ is also drawn from $D^o_{XY}$. Thus, $D^1_{Y|X} = D^o_{Y|X}$.

Then we prove the cases when $t > 1$ using mathematical induction as follows:

1. Base case: We know $\mathcal{S}_2 = \mathcal{S}_1 \cup \mathcal{S}_{o,2} \cup \mathcal{S}_{m,1}$. The first two sets on rhs are drawn from $D^o_{XY}$. The labeing in the third set is produced by $h_1(x) = D^1_{Y|X} = D^o_{Y|X}$. Thus, the base case is proved.

2. Induction step: $\mathcal{S}_t = \mathcal{S}_{t-1} \cup \mathcal{S}_{o,t} \cup \mathcal{S}_{m,t-1}$. Similarly, we only need to consider the third set. But Note that $h_t(x) = D^{t-1}_{Y|X} = D^o_{Y|X}$, the induction step is easily completed. $\qquad\square$

## G.8 Proof of Proposition G.3

At round $t$, we know $A^{it}_{\theta^i_t} > A^{jt}_{\theta^j_t}$. To reach demographic parity, the acceptance rates need to be the same, so at least one of the following situations must happen: (i) $\widetilde{\theta}^i_t > \theta^i_t$ and $\widetilde{\theta}^j_t > \theta^j_t$; (ii) $\widetilde{\theta}^i_t < \theta^i_t$ and $\widetilde{\theta}^j_t < \theta^j_t$; (iii) $\widetilde{\theta}^i_t \geq \theta^i_t$ and $\widetilde{\theta}^j_t \leq \theta^j_t$. Next we prove that (i) and (ii) cannot be true by contradiction.

Suppose (i) holds, then we can find $\overline{\theta}^j_t < \theta^j_t$ such that $\ell(\overline{\theta}^j_t, \mathcal{S}^j_t) \in \left(\ell(\theta^j_t, \mathcal{S}^j_t), \ell(\widetilde{\theta}^j_t, \mathcal{S}^j_t)\right)$ and $A^{jt}_{\overline{\theta}^j_t} \in \left(A^{jt}_{\theta^j_t}, A^{it}_{\theta^i_t}\right)$. We can indeed find this $\overline{\theta}^j_t$ because $\ell, A$ are continuous w.r.t. $\theta$. Now noticing that $A^{it}_{\overline{\theta}^i_t} = A^{jt}_{\overline{\theta}^j_t} > A^{jt}_{\theta^j_t}$ but $A^{jt}_{\overline{\theta}^j_t} < A^{jt}_{\theta^j_t}$, we will know we can find $\overline{\theta}^i_t \in \left(\theta^i_t, \widetilde{\theta}^i_t\right)$ to satisfy demographic parity together with $\overline{\theta}^j_t$. Since $P_{Y|X}(1|x)$ satisfies monotonic likelihood and $\theta^i_t$ is the optimal point, $\ell(\overline{\theta}^i_t, \mathcal{S}^i_t) < \ell(\widetilde{\theta}^i_t, \mathcal{S}^i_t)$ must hold. Thus, $(\overline{\theta}^i_t, \overline{\theta}^j_t)$ satisfy demographic parity and have a lower loss than $(\widetilde{\theta}^i_t, \widetilde{\theta}^j_t)$. Moreover, the pair satisfies (iii), which produces a conflict.

Similarly, we can prove (ii) cannot hold by contradiction, thereby proving (iii) must hold.

## G.9 Proof of Theorem 4.1

Situation (i) is directly derived from Thm. 3.3, where we can just regard the trajectory of the acceptance rate of group $j$ as moving the one of group $i$ vertically downward; Situation (ii) is derived from Prop. D.2, where the systematic bias causes unfairness and it keeps stable; Situation (iii) can be derived from Thm. 3.6, where the acceptance rate of group $i$ stays stable, while the one for $j$ monotonically increases.

## G.10 Proof of Theorem 4.2

The first fact is that both $\theta^i_t, \theta^j_t$ stay stable as $0.5 - \mu_i$ and $0.5 - \mu_j$ when we use accuracy as the metric. Therefore, to understand which group is disadvantaged/advantaged at $t + 1$, we just need to track the agents who best respond to $\widetilde{\theta}^i_t, \widetilde{\theta}^j_t$. First, we need the following proposition which is proved in App. G.8:

**Proposition G.3.** *If $P^{ti}_{XY}, P^{tj}_{XY}$ are continuous, then $\widetilde{\theta}^i_t \geq \theta^i_t$ and $\widetilde{\theta}^j_t \leq \theta^j_t$.*

Note that we want to sacrifice the least accuracy to reach demographic parity. Since the agent best response will never make them worse off, we will know $a_t^i$ corresponding to the acceptance rate of group $i$ under the optimal threshold $\theta_t^i$ is always larger than or equal to $a_0^i$. Thus, it suffices to show that under the condition of Thm. 4.2, $a_t^j < a_0^i$.

Suppose $a_t^j < a_0^i$, then we can use Chebyshev's Inequality to bound the probability that an unadmitted agent $z$ in group $j$ reaches $\theta_t^j$ after its best response. Denote this event as $E$ and the threshold agent $z$ feels is $I_z$, where $E(I_z) = \widetilde{\theta_t^j}$. Then we can write $\mathbb{P}(E) < \mathbb{P}(I_z - E(I_z) \geq \theta_t^j - \widetilde{\theta_t^j}) < \frac{(\theta_t^j - \widetilde{\theta_t^j})^2}{\sigma_t^2}$. Thus, when $\sigma_t < \frac{\theta_t^j - \widetilde{\theta_t^j}}{\sqrt{a_0^i - a_t^j}}$, we know $\mathbb{P}(E) < a_0^i - a_t^j$, and from the simple fact $a_{t+1}^j < a_t^j + \mathbb{P}(E)$, we know $j$ will stay disadvantaged at $t + 1$.

However, when we do not add the fairness intervention, just consider the situation where all unadmitted agents in group $j$ at $t = 0$ improve, while no one in group $i$ can improve. This will flip the disadvantaged/advantaged group.

# H All plots with error bars

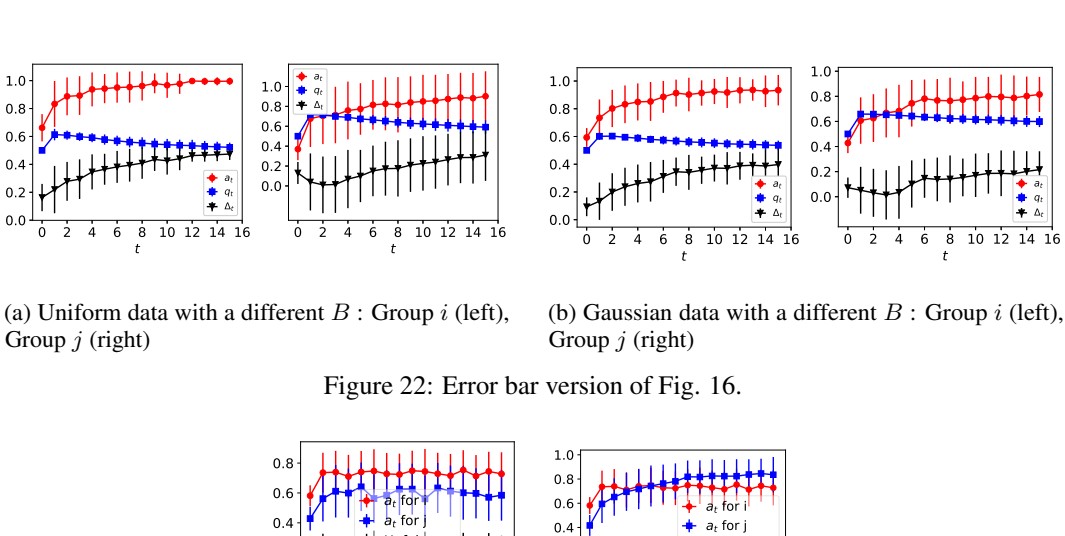

(a) Uniform data with a different $B$ : Group $i$ (left), Group $j$ (right)

(b) Gaussian data with a different $B$ : Group $i$ (left), Group $j$ (right)

Figure 22: Error bar version of Fig. 16.

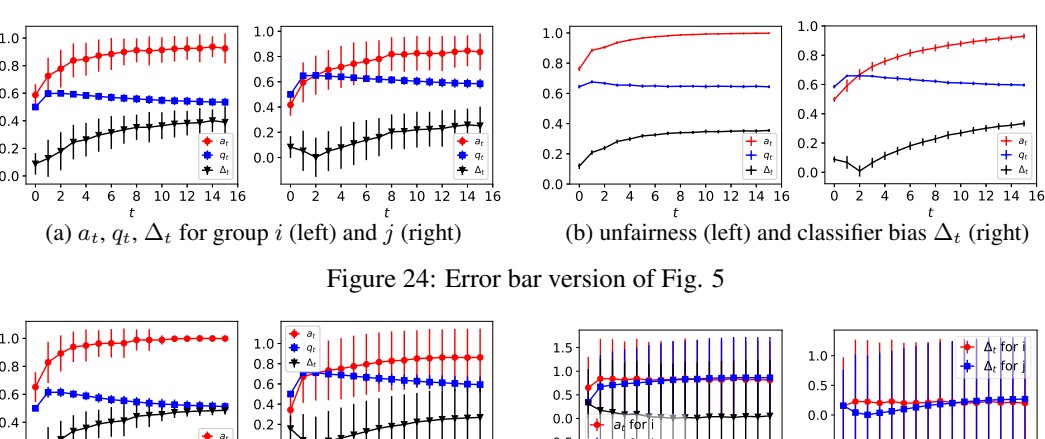

Figure 23: Error bar version of Fig. 4

(a) $a_t, q_t, \Delta_t$ for group $i$ (left) and $j$ (right)

(b) unfairness (left) and classifier bias $\Delta_t$ (right)

Figure 24: Error bar version of Fig. 5

(a) $a_t, q_t, \Delta_t$ for group $i$ (left) and $j$ (right)

(b) unfairness (left) and classifier bias $\Delta_t$ (right)

Figure 25: Error bar version of Fig. 9

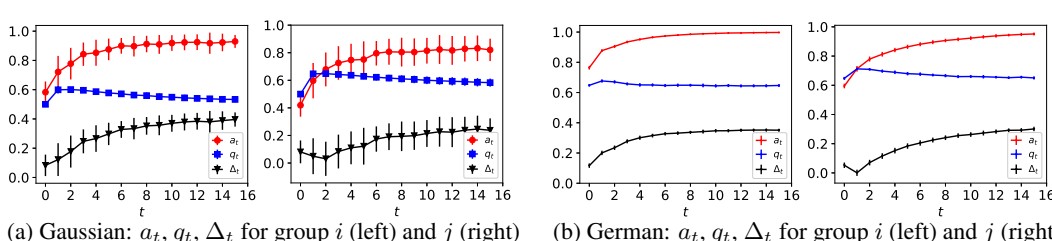

(a) Gaussian: $a_t, q_t, \Delta_t$ for group $i$ (left) and $j$ (right)

(b) German: $a_t, q_t, \Delta_t$ for group $i$ (left) and $j$ (right)

Figure 26: Error bar version of Fig. 6

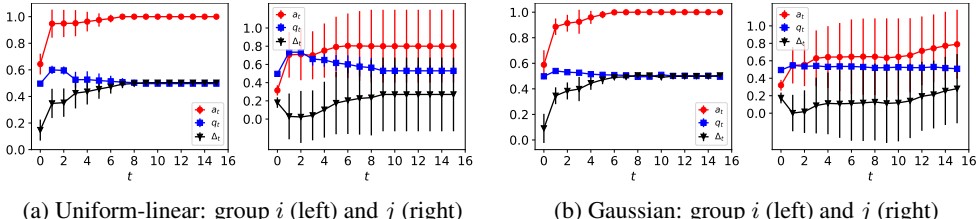

(a) Uniform-linear: group $i$ (left) and $j$ (right)    (b) Gaussian: group $i$ (left) and $j$ (right)

Figure 27: Error bar version of Fig. 8.

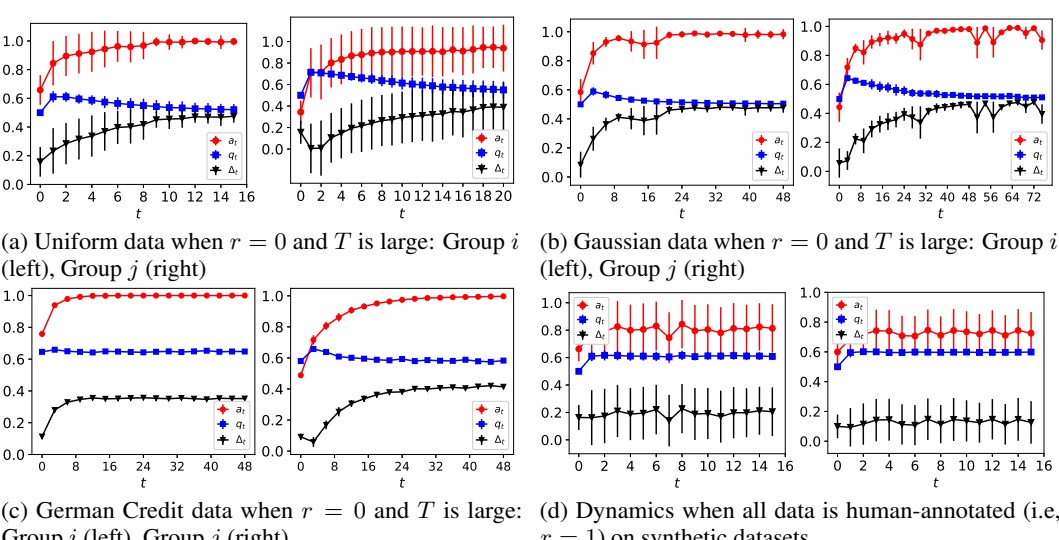

(a) Uniform data when $r = 0$ and $T$ is large: Group $i$ (left), Group $j$ (right)

(b) Gaussian data when $r = 0$ and $T$ is large: Group $i$ (left), Group $j$ (right)

(c) German Credit data when $r = 0$ and $T$ is large: Group $i$ (left), Group $j$ (right)

(d) Dynamics when all data is human-annotated (i.e, $r = 1$) on synthetic datasets.

Figure 28: Error bar version of Fig. 15

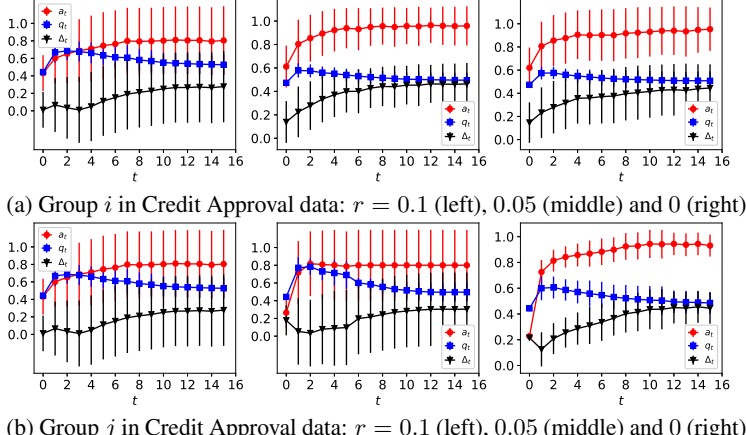

(a) Group $i$ in Credit Approval data: $r = 0.1$ (left), 0.05 (middle) and 0 (right)

(b) Group $j$ in Credit Approval data: $r = 0.1$ (left), 0.05 (middle) and 0 (right)

Figure 29: Error bar version of Fig. 11

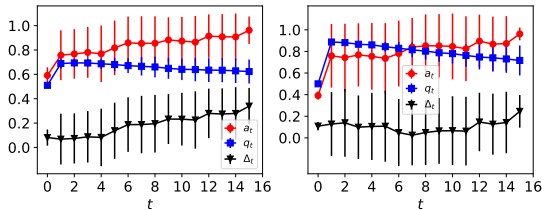

Figure 30: Error bar version of Fig. 14.

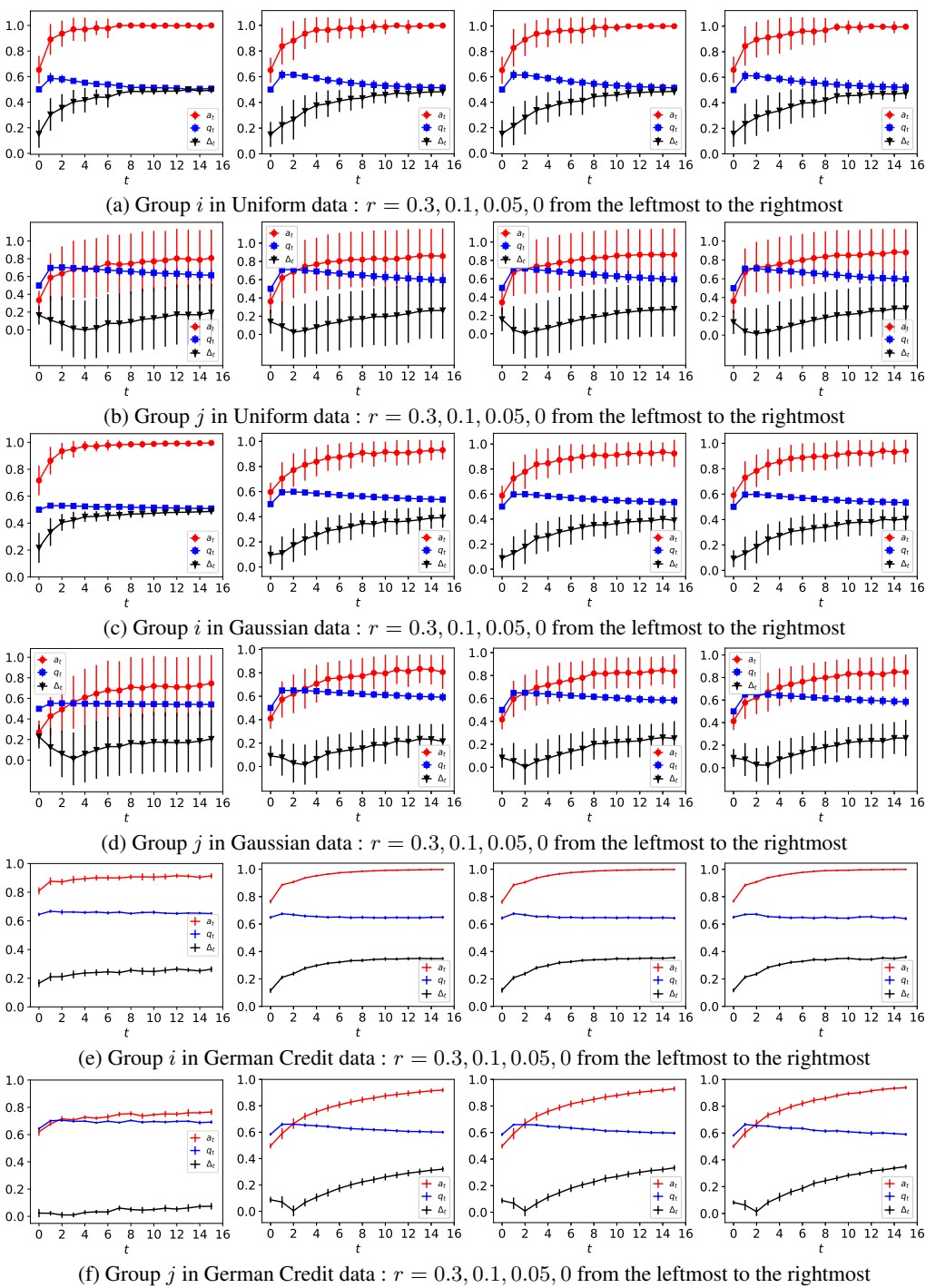

(a) Group $i$ in Uniform data : $r = 0.3, 0.1, 0.05, 0$ from the leftmost to the rightmost

(b) Group $j$ in Uniform data : $r = 0.3, 0.1, 0.05, 0$ from the leftmost to the rightmost

(c) Group $i$ in Gaussian data : $r = 0.3, 0.1, 0.05, 0$ from the leftmost to the rightmost

(d) Group $j$ in Gaussian data : $r = 0.3, 0.1, 0.05, 0$ from the leftmost to the rightmost

(e) Group $i$ in German Credit data : $r = 0.3, 0.1, 0.05, 0$ from the leftmost to the rightmost

(f) Group $j$ in German Credit data : $r = 0.3, 0.1, 0.05, 0$ from the leftmost to the rightmost

Figure 31: Error bar version of Fig. 17.

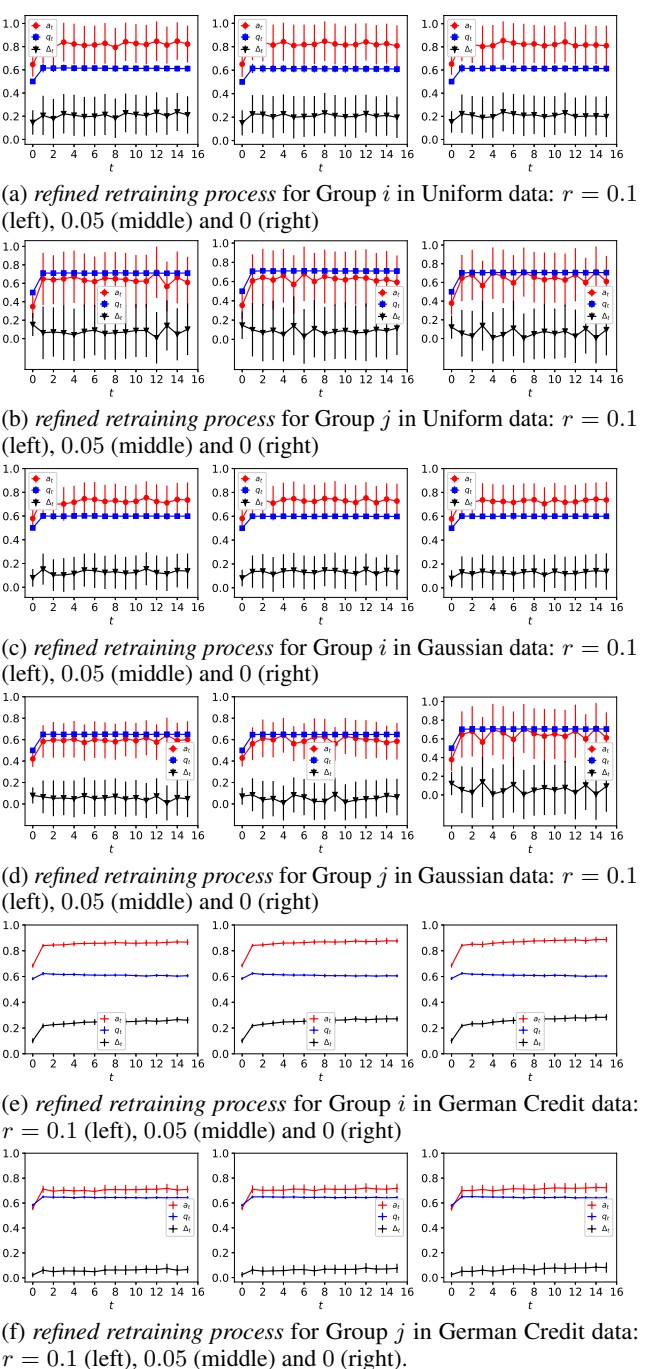

(a) *refined retraining process* for Group $i$ in Uniform data: $r = 0.1$ (left), 0.05 (middle) and 0 (right)

(b) *refined retraining process* for Group $j$ in Uniform data: $r = 0.1$ (left), 0.05 (middle) and 0 (right)

(c) *refined retraining process* for Group $i$ in Gaussian data: $r = 0.1$ (left), 0.05 (middle) and 0 (right)

(d) *refined retraining process* for Group $j$ in Gaussian data: $r = 0.1$ (left), 0.05 (middle) and 0 (right)

(e) *refined retraining process* for Group $i$ in German Credit data: $r = 0.1$ (left), 0.05 (middle) and 0 (right)

(f) *refined retraining process* for Group $j$ in German Credit data: $r = 0.1$ (left), 0.05 (middle) and 0 (right).

Figure 32: Error bar version of Fig. 18

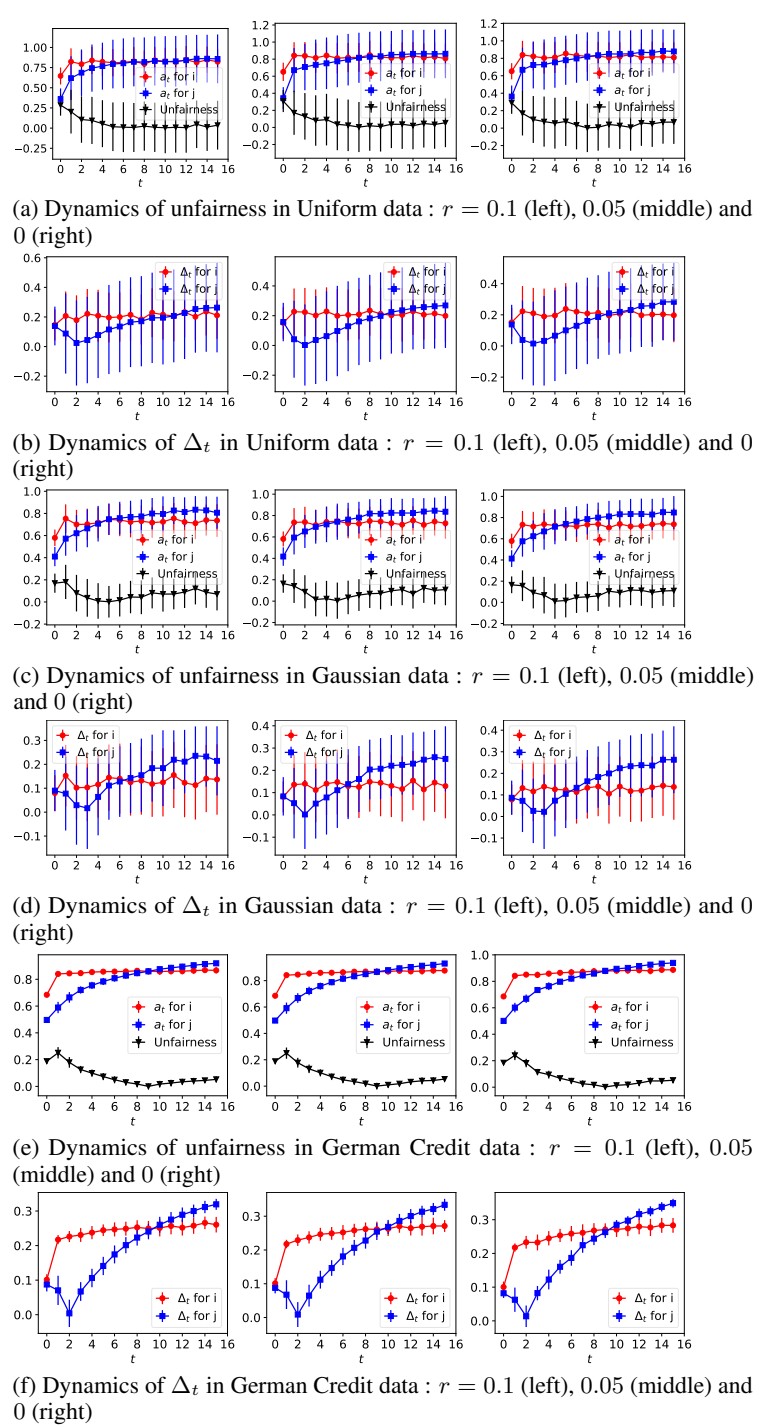

(a) Dynamics of unfairness in Uniform data : $r = 0.1$ (left), $0.05$ (middle) and $0$ (right)

(b) Dynamics of $\Delta_t$ in Uniform data : $r = 0.1$ (left), $0.05$ (middle) and $0$ (right)

(c) Dynamics of unfairness in Gaussian data : $r = 0.1$ (left), $0.05$ (middle) and $0$ (right)

(d) Dynamics of $\Delta_t$ in Gaussian data : $r = 0.1$ (left), $0.05$ (middle) and $0$ (right)

(e) Dynamics of unfairness in German Credit data : $r = 0.1$ (left), $0.05$ (middle) and $0$ (right)

(f) Dynamics of $\Delta_t$ in German Credit data : $r = 0.1$ (left), $0.05$ (middle) and $0$ (right)

Figure 33: Error bar version of Fig. 19

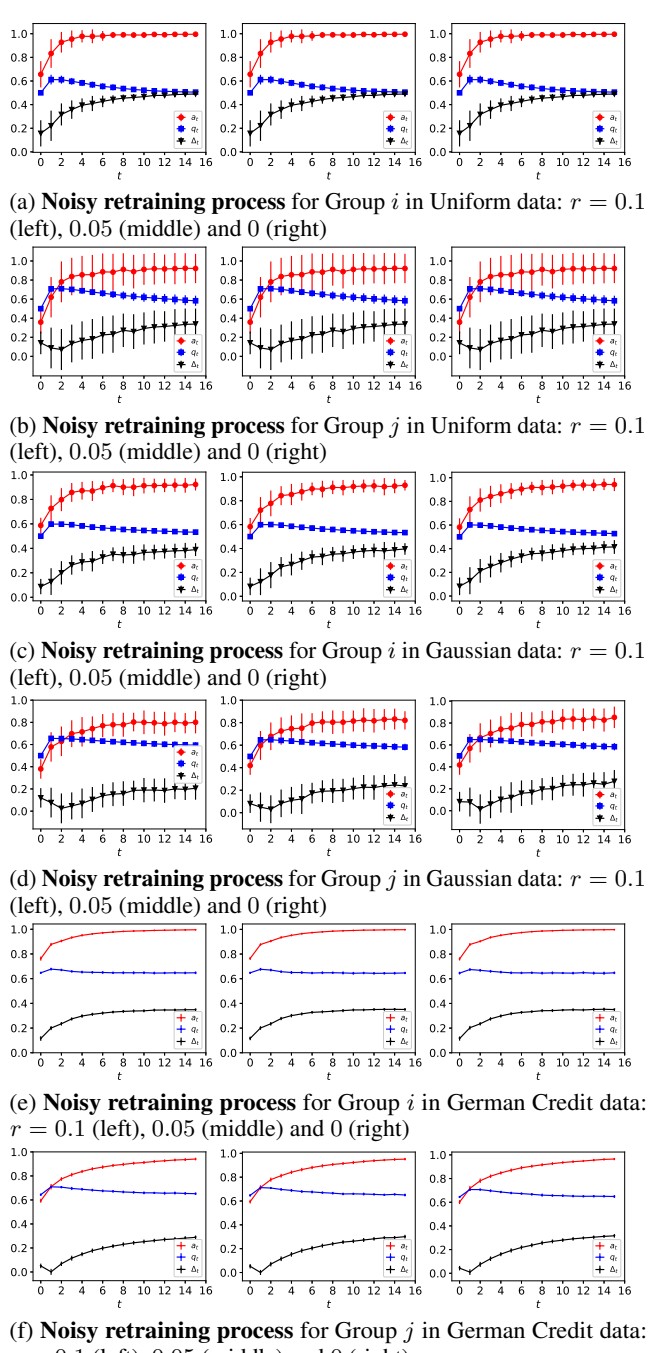

(a) **Noisy retraining process** for Group $i$ in Uniform data: $r = 0.1$ (left), 0.05 (middle) and 0 (right)

(b) **Noisy retraining process** for Group $j$ in Uniform data: $r = 0.1$ (left), 0.05 (middle) and 0 (right)

(c) **Noisy retraining process** for Group $i$ in Gaussian data: $r = 0.1$ (left), 0.05 (middle) and 0 (right)

(d) **Noisy retraining process** for Group $j$ in Gaussian data: $r = 0.1$ (left), 0.05 (middle) and 0 (right)

(e) **Noisy retraining process** for Group $i$ in German Credit data: $r = 0.1$ (left), 0.05 (middle) and 0 (right)

(f) **Noisy retraining process** for Group $j$ in German Credit data: $r = 0.1$ (left), 0.05 (middle) and 0 (right)

Figure 34: Error bar version of Fig. 20

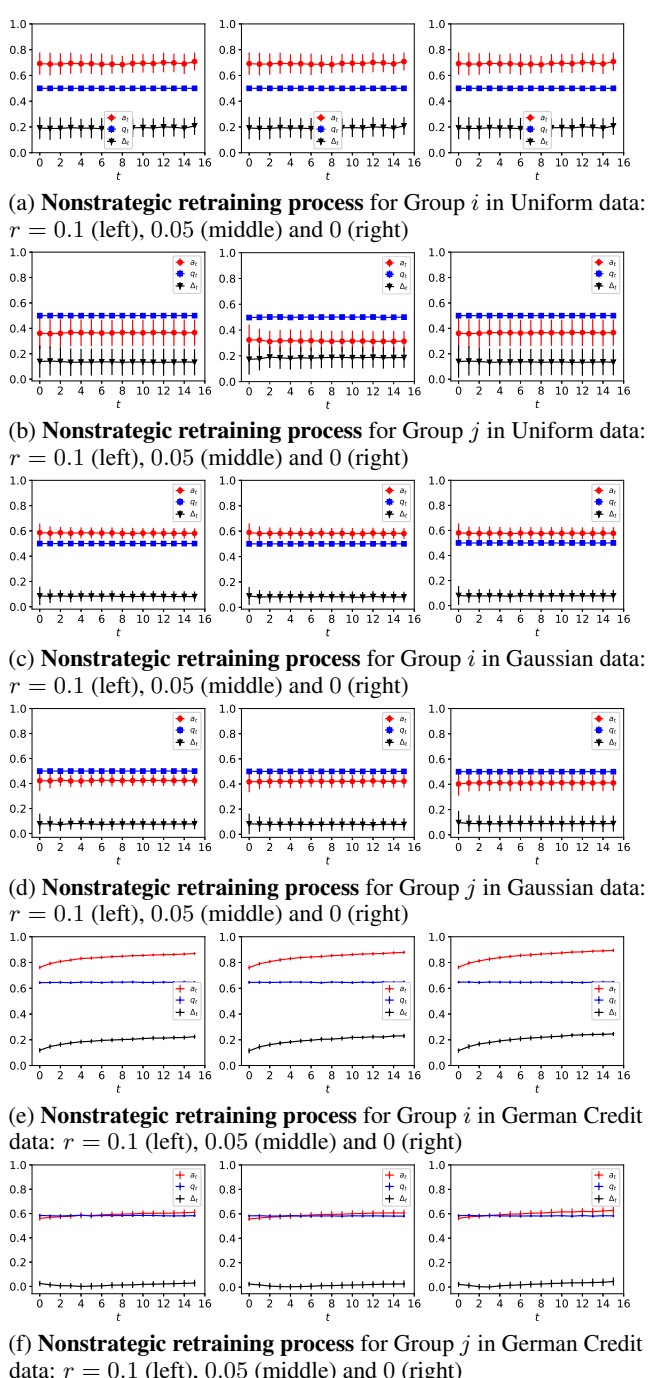

(a) **Nonstrategic retraining process** for Group $i$ in Uniform data: $r = 0.1$ (left), 0.05 (middle) and 0 (right)

(b) **Nonstrategic retraining process** for Group $j$ in Uniform data: $r = 0.1$ (left), 0.05 (middle) and 0 (right)

(c) **Nonstrategic retraining process** for Group $i$ in Gaussian data: $r = 0.1$ (left), 0.05 (middle) and 0 (right)

(d) **Nonstrategic retraining process** for Group $j$ in Gaussian data: $r = 0.1$ (left), 0.05 (middle) and 0 (right)

(e) **Nonstrategic retraining process** for Group $i$ in German Credit data: $r = 0.1$ (left), 0.05 (middle) and 0 (right)

(f) **Nonstrategic retraining process** for Group $j$ in German Credit data: $r = 0.1$ (left), 0.05 (middle) and 0 (right)

Figure 35: Error bar version of Fig. 21

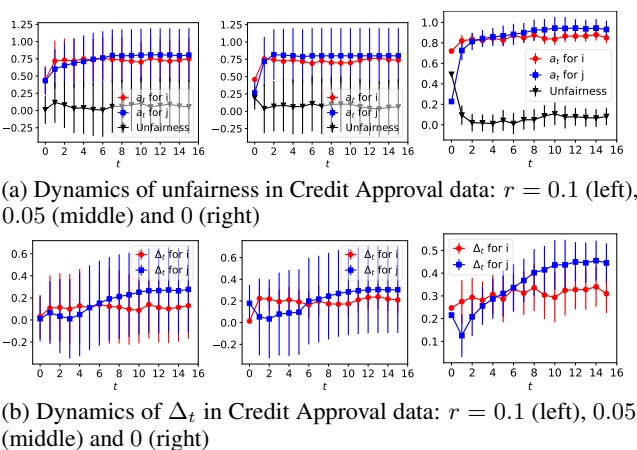

(a) Dynamics of unfairness in Credit Approval data: $r = 0.1$ (left), 0.05 (middle) and 0 (right)

(b) Dynamics of $\Delta_t$ in Credit Approval data: $r = 0.1$ (left), 0.05 (middle) and 0 (right)

Figure 36: Error bar version of Fig. 12

Although all our theoretical results are expressed in terms of expectation, we provide error bars for all plots in the main paper, App. E and F if randomness is applicable. All the above figures demonstrate expectations as well as error bars ($\pm$ 1 standard deviation). Overall, the experiments have reasonable standard errors. However, experiments in the Credit Approval dataset [20] (Fig. 36) incur larger standard errors, which is not surprising because the dataset violates several assumptions. Finally, note that we conduct 50-100 randomized trials for every experiment and we should expect much lower standard errors if the numbers of trials become large.

