# OpenReview forum: "Automating Data Annotation under Strategic Human Agents: Risks and Potential Solutions"
_NeurIPS.cc/2024/Conference — NeurIPS 2024 poster_

### Official Review · Reviewer_ynjw · 2024-07-11

**Soundness:** 3
**Presentation:** 3
**Contribution:** 3
**Rating:** 5
**Confidence:** 3

**Summary:**

This paper investigates an intriguing problem regarding how machine learning models evolve during dynamic retraining using model-annotated samples, incorporating strategic human responses. The authors discover that it becomes increasingly likely for individuals to receive positive decisions as the model undergoes retraining, although the proportion of individuals with positive labels may decrease over time. To stabilize the dynamics, the authors propose a refined retraining process. They also examine how these retraining processes can impact algorithmic fairness and find that enforcing common fairness constraints in every retraining round may not benefit the disadvantaged groups in the long term. Experiments conducted on both synthetic and real-world data validate the findings of this study.

**Strengths:**

1.	The paper thoroughly analyzes how humans, acting as strategic agents, adapt their behavior in response to ML systems and how this behavior impacts the retraining process of ML systems. By formalizing these interactions and analyzing their long-term dynamics, the paper provides a theoretical foundation for understanding and predicting these complex interactions. This in-depth analysis helps uncover potential systemic issues and offers concrete theoretical support for improving models.
2.	The paper not only identifies potential risks of retraining ML models with model-annotated data but also proposes an improved retraining method using a probabilistic sampler to enhance the quality of model-annotated samples. This method aims to stabilize the dynamics of acceptance and qualification rates, reducing classifier bias. The proposed solution is innovative and practical, helping to mitigate negative impacts in real-world applications.
3.	The paper combines theoretical analysis with experiments on semi-synthetic and real data to validate the findings. The experimental results, which show consistent dynamics with theoretical predictions, enhance the credibility and applicability of the research. This approach ensures the reliability of the research findings by providing both theoretical and empirical support.

**Weaknesses:**

1.	While the conclusions of the paper help understand the impact of human strategic behavior on ML systems, many of these conclusions are somewhat intuitive and straightforward. For instance, the increase in acceptance rates and the potential decrease in qualification rates over time are logically reasonable but do not provide particularly new insights. Delving deeper into underlying mechanisms or revealing more complex interactions could make the research more innovative and impactful.
2.	The scale of the datasets used in experiments is relatively small, which may not fully capture the complexity of system dynamics in large-scale data environments.
3.	The paper primarily focuses on linear models and specific distributions. The research conclusions might not fully apply to non-linear models and complex data distributions.

**Questions:**

See weaknesses

**Limitations:**

N.A

---

> ### Author Rebuttal · Authors · 2024-08-05
>
> Thanks for the comments. We address your questions point by point as follows.
>
> > While the conclusions of the paper help understand the impact of human strategic behavior on ML systems, many of these conclusions are somewhat intuitive and straightforward.
>
> We believe the theoretical results are not intuitive. In section 3, we show that although the decision-maker keeps retraining the classifier, the increasing $a_t$ and decreasing $q_t$ demonstrate the model becomes more biased (Theorem 3.3 - 3.5). In section 4, we not only display the fairness dynamics of the retraining process but also research the influence of fairness interventions. In particular, Theorem 4.2 reveals that fairness intervention can even prohibit the disadvantaged group from becoming advantaged, which is also counter-intuitive.
>
>
> > The paper primarily focuses on linear models and specific distributions. The research conclusions might not fully apply to non-linear models and complex data distributions.
>
> - Note that the linearity of the decision model is standard setting in strategic classification (e.g., [1,6,7,11,12,17,25,31,35] in the main paper). In practice, one can first use a non-linear feature extractor to learn the embeddings of agents' preliminary features and then apply a linear model to the newly generated embedded features. Given the knowledge of both feature extractor and model, agents will best respond with respect to new features, and all our results still hold. The generalization to non-linear settings is also discussed in [13,28] in the main paper and [Levanon and Rosenfeld, 2021].
>
> - To further validate the above arguments, we provide an additional case study using ACSIncome data [Ding et al., 2022], which contains information on more than $150K$ agents. In this study, the goal is to predict whether a person has an annual income $> 50000$ based on $53$ features such as education level and working hours per week. Specifically, we consider a decision-maker who first learns 2-D embeddings from $53$ original features using a neural network and then regards the embedding as the new feature. A linear decision model is trained and used on these new features to make predictions. We divide the agents into 2 groups based on their ages. Similar to the credit approval data, we then fit Beta distributions on the 2 groups and then verify the monotonic likelihood assumption (Figure 1,2 in the attached pdf). We then plot the dynamics of $a_t, q_t, \Delta_t$ for both groups when the systematic bias is either positive or negative. The results show that similar trends still hold for this large dataset (Figure 3 in the attached pdf).
>
> - Finally, it is an interesting direction to extend our setting into non-linear decision policy while the agents can only respond to the classifier with the original features (not the embedding). This setting can be highly intricate because both the decision boundary and the agent best response will vary. We hope our paper can be a starting point and a solid foundation of more future work.
>
> > The scale of the datasets used in experiments is relatively small, which may not fully capture the complexity of system dynamics in large-scale data environments.
>
> -  Our paper focuses on strategic classification settings [1,6,7,11,12,17,25,31,35]. According to the previous literature, human strategic behaviors are prevalent in high-stakes domains such as hiring, lending, and college admission, where humans have more incentives to improve their features for favorable outcomes. We believe the datasets adopted in our experiments are sufficiently representative and cover well-known data widely used in strategic classification literature.
>
> - The additional case study illustrated in the above answer uses a larger dataset.
>
>
> ### References
>
> Ding, F., Hardt, M., Miller, J., & Schmidt, L. (2021). Retiring adult: New datasets for fair machine learning. Advances in neural information processing systems, 34, 6478-6490.

---

### Official Review · Reviewer_JSvV · 2024-07-12

**Soundness:** 4
**Presentation:** 2
**Contribution:** 3
**Rating:** 5
**Confidence:** 4

**Summary:**

This paper addresses the dynamics of machine learning systems when retrained with model-annotated and human-annotated samples in the presence of strategic human agents. It explores how these dynamics affect various welfare aspects, including those of applicants, decision-makers, and the broader social context. The work emphasizes the potential risks and unintended consequences of retraining classifiers in strategic settings, emphasizing the complex interplay between agent strategies and ML system updates.

**Strengths:**

The paper addresses an underexplored aspect of machine learning: the interaction of strategic human agents with ML systems over iterative retraining cycles. The incorporation of both model-annotated and human-annotated data into the retraining process, alongside the strategic adaptations of agents, presents a novel problem formulation.

**Weaknesses:**

1. The theoretical results depend on assumptions that may not hold in more complex or varied real-world scenarios, and the Gaussian, German Credit, and Credit Approval datasets in the experiment violate some conditions in the theoretical analysis. This limitation may affect the generalizability of the results.
2. It would be beneficial to organize the experiments section of the main paper, providing a more detailed display of results related to the refined retraining process, which is a main contribution of this work.
3. The study primarily examines binary classification scenarios with only two outcome classes (e.g., admitted/not admitted). It would be beneficial if the authors could explore how their findings and methodologies might be applicable to more complex machine learning tasks, such as multi-class classification or ranking problems, which are more common in real-world settings.

**Questions:**

Please see the weaknesses part.

**Limitations:**

The authors have adequately addressed the limitations and societal impact.

---

> ### Author Rebuttal · Authors · 2024-08-05
>
> Thanks for the comments. We address your questions point by point as follows.
>
> > Assumptions of the theoretical results and the generalizability
>
> - Our paper focuses on strategic classification settings [1,6,7,11,12,17,25,31,35]. According to the previous literature, human strategic behaviors are prevalent in high-stakes domains such as hiring, lending, and college admission, where humans have more incentives to improve their features for favorable outcomes. We believe the datasets adopted in our experiments are sufficiently representative and cover well-known data widely used in strategic classification literature. While the Gaussian, German Credit, and Credit Approval datasets violate some conditions, the experimental results still show similar trends as the theoretical results, demonstrating the robustness and practical value of the theory.
>
> - The linearity assumption of the decision model is standard setting in strategic classification. In practice, one can first use a non-linear feature extractor to learn the embeddings of agents' preliminary features and then apply a linear model to the newly generated embedded features. Given the knowledge of both feature extractor and model, agents will best respond with respect to new features and all our results still hold. The generalization to non-linear settings is also discussed in [13,28] in the main paper and [Levanon and Rosenfeld, 2021].
>
> - To further validate the above arguments, we provide an additional case study using ACSIncome data [Ding et al., 2022], which contains information on more than $150K$ agents. In this study, the goal is to predict whether a person has an annual income $> 50000$ based on $53$ features such as education level and working hours per week. Specifically, we consider a decision-maker who first learns 2-D embeddings from $53$ original features using a neural network and then regards the embedding as the new feature. A linear decision model is trained and used on these new features to make predictions. We divide the agents into 2 groups based on their ages. Similar to the credit approval data, we then fit Beta distributions on the 2 groups and then verify the monotonic likelihood assumption (Figure 1,2 in the attached pdf). We then plot the dynamics of $a_t, q_t, \Delta_t$ for both groups when the systematic bias is either positive or negative. The results show that similar trends still hold for this large dataset (Figure 3 in the attached pdf).
>
> > Organize the experiments section of the main paper
>
> Thanks for the suggestion. We will move the content of App F.3 to the main paper.
>
> > The study primarily examines binary classification scenarios with only two outcome classes (e.g., admitted/not admitted).
>
> - Similar to most studies in strategic classification [1,6,7,11,12,17,25,31,35], we focus on binary classification which is a standard setting in strategic classification and has many real applications (e.g., hiring, admission, loan approval) that involve binary decision-making on humans. Compared to existing works, ours is the first that examines the long-term impact of automating data annotation under strategic human agents.
>
> - Although the theoretical modeling and analysis under binary classification is already non-trivial, we may extend the analytical framework and insights to multi-class classification. Specifically, consider strategic agents with categorical labels $Y\in \mathcal{Y}$. Instead of improving features toward one specific favorable outcome (e.g., acceptance in lending/hiring, as given in Eq. (1)), each agent may have its own target $y^*\in \mathcal{Y}$ and they may best respond based on $x_t = argmax_z {Pr(f_{t-1}(z)=y^*)-c(z,x)}$. For the decision-maker, the model retraining process remains the same, i.e., it augments training data with model-annotated samples and human-annotated samples at every round. Under this broader setting, we may consider two scenarios: (i) all agents have the same target $y^*$; (ii) agents have different target $y^*$. For case (i), because agents move toward one class, the result is similar to a binary setting: instead of having the acceptance rate (resp. qualification rate) increasing (resp. decreasing) over time, the probability $P(f_t(X)=y^*)$ (resp. $P(Y=y^*)$) increases (resp. decreases) under certain conditions. For case (ii), the results will be more complicated because different targets could induce highly diverse agent behavior that disrupts the monotonicity change of distributions $P(f_t(X))$ and $P(Y)$. This is an interesting future direction.
>
> - Last, we agree with the reviewer that it is interesting to extend our work to multi-class classification or ranking, which we hope to study in the future. Indeed, such extension is non-trivial and has been an ongoing effort of the community. For example, [Liu et al., 2022] consider competition among strategic agents and is the first work studying strategic ranking to the best of our knowledge. We hope our paper can provide insights and shed light on future works.
>
> ### References
>
> Ding, F., Hardt, M., Miller, J., & Schmidt, L. (2021). Retiring adult: New datasets for fair machine learning. Advances in neural information processing systems, 34, 6478-6490.
>
> Liu, Lydia T., Nikhil Garg, and Christian Borgs. "Strategic ranking." International Conference on Artificial Intelligence and Statistics. PMLR, 2022.
>
> Levanon, S., & Rosenfeld, N. (2021, July). Strategic classification made practical. In International Conference on Machine Learning (pp. 6243-6253). PMLR.

---

> > ### Comment · Reviewer_JSvV · 2024-08-12
> >
> > Thank you for your response and the efforts made to address the initial concerns. We appreciate the clarifications and the additional experiments you provided. However, some concerns remain regarding the alignment between the theoretical assumptions and the empirical data, especially with datasets that do not fit these assumptions. While the trends appear consistent, this misalignment could limit the practical applicability of the theoretical outcomes. Additionally, we would benefit from a more detailed exploration of the specific challenges encountered when adapting your theoretical results to more complex class distributions to fully appreciate the breadth and impact of your contributions. Therefore, I will maintain my current score.

---

### Official Review · Reviewer_ovmu · 2024-07-12

**Soundness:** 3
**Presentation:** 3
**Contribution:** 2
**Rating:** 5
**Confidence:** 3

**Summary:**

The paper explores a scenario where a machine learning system retrains itself over time by collecting data generated through ML system annotation itself as well as human-annotated data while allowing the distribution of training samples to evolve over time. This evolution is influenced by the strategic behavior of samples that may be marked as positive by the system. Based on this model, the paper investigates the dynamic processes of several metrics, including acceptance rate, qualification, and classifier bias.

**Strengths:**

1. The problem is well-motivated.

2. The high-level model construction is interesting. For instance, the model considers the sequential aspects of the problem and analyzes how certain metrics change over time, which is more realistic than previous works.

3. Additionally, the author incorporates 'systematic bias' into the modeling and discusses numerous real-world examples related to algorithmic fairness, an important topic for the community.

**Weaknesses:**

1. Some aspects of the results seem trivial or straightforward given the assumptions in the model. (Please see my questions below.)

2. The writing could be improved, as some statements in the paper are confusing. (Please see my questions below.)

3. The figure size, legend, and overall organization could be improved. For example, in Figure 2, the blue and red dots in the legend are too small, making the figure difficult to interpret. Additionally, the y-axis legend in Figure 2 uses different scales, complicating comparisons.

**Questions:**

1. Assuming $N \gg K$ and that strategic agents, using the output of the previous classifier, adapt to the classifier's outcome, it seems straightforward that the acceptance rate should increase. Could you clarify what the non-trivial or surprising aspect of Theorem 3.3 is?

2. To what extent does the conclusion/results in Theorem 3.5 hold for classifiers beyond linear ones?

3. In what situations can the convexity assumption in Theorem 3.5 be validated in practice?

4. What does the term *cumulative* density function exactly refer to in the statement of Theorem 3.5? The domain of a standard cumulative function should be $\mathbb{R}$ based on the usual definition. How, then, can it be restricted to a half-space $J \subset \mathbb{R}^d$ in Theorem 3.5? Is this a typo, or am I missing something?

5. Line 99: What does the continuity of $P(Y | X)$ means here when $Y$ takes values in $\{0, 1\}$?

6. Lines 187-188: Why does $\mathcal{S}{m,t-1}$ have a higher qualification rate than $ \mathcal{S}{t-1} $? Shouldn't that be the acceptance rate? Or am I missing something?

**Limitations:**

The authors have discussed the limitations of their work.

---

> ### Author Rebuttal · Authors · 2024-08-05
>
> Thanks for the comments. We address your questions point by point as follows.
>
> > Theorem 3.3
>
> The result in Theorem 3.3 (i.e., $a_t > a_{t-1}$) does **not** require $N >> K$ but only needs $N > 0$. Condition $N >> K$ is only used in Proposition 3.4, under which the acceptance rate converges to $1$ as $t \to \infty$ (i.e., all agents are admitted in the long run). For general cases with arbitrary positive $N$, we believe the increasing trend of the acceptance rate (Theorem 3.3) under complex dynamics between retrained model and strategic agents is not straightforward, rigorously proving requires non-trivial efforts by induction (see detailed proof in Appendix).
>
> > Theorem 3.5 beyond linear classifiers?
>
> - Note that the linearity of the decision model is standard setting in strategic classification (e.g., [1,6,7,11,12,17,25,31,35]). In practice, one can first use a non-linear feature extractor to learn the embeddings of agents' preliminary features and then apply a linear model to the newly generated embedded features. Given the knowledge of both feature extractor and model, agents will best respond with respect to new features, and all our results still hold. The generalization to non-linear settings is also discussed in [13,28] in the main paper and [Levanon and Rosenfeld, 2021].
>
> - To further validate the above arguments, we provide an additional case study using ACSIncome data [Ding et al., 2022] with information on more than $150K$ agents. The goal is to predict whether a person has an annual income $> 50000$ based on $53$ features such as education level and working hours. Specifically, we consider a decision-maker who first learns 2-D embeddings from $53$ original features using a neural network and then regards the embedding as the new feature. A linear decision model is trained and used on these new features to make predictions. We divide the agents into 2 groups based on their ages. Similar to the credit approval data, we then fit Beta distributions on the 2 groups and then verify the monotonic likelihood assumption (Figure 1,2 in the attached pdf). We then plot the dynamics of $a_t, q_t, \Delta_t$ for both groups when the systematic bias is either positive or negative. The results show that similar trends still hold for this large dataset (Figure 3 in the attached pdf).
>
> > In what situations can the convexity assumption in Theorem 3.5 be validated in practice?
>
> - In lines 236-238 we discussed the situations where the conditions of Theorem 3.5 hold theoretically (e.g. when $P_X$ belongs to Uniform distribution, Beta distribution, or Gaussian distribution when the first decision policy admits 50\% or more agents). In practice, the decision-maker can fit the empirical distribution of the real data to get $F_X$ and then verify the convexity. Particularly, if the cdf does not have analytical form, we can verify the convexity empirically by sampling the points from the domain and check the convexity inequality $\forall x_1, x_2 \in J , \lambda \in (0,1), f(\lambda x_1 + (1 - \lambda) x_2) \leq \lambda f(x_1) + (1 - \lambda) f(x_2)$.
>
> - The assumptions are sufficient but not necessary, i.e., Theorem 3.5 may still hold even when assumptions are violated. This is mentioned in lines 238-239 and validated in experiments. Among all datasets adopted in experiments, only the "uniform-linear" dataset satisfies all assumptions in Theorem 3.5, while the Gaussian dataset satisfies Theorem 3.5 only if the initial qualification rate of agents is larger than 0.5. The German Credit and Credit Approval dataset only satisfies the linear classifier assumption and the monotonic likelihood assumption. However, the empirical results on all datasets demonstrate the validity of Theorem 3.5. Thus, we believe the phenomenon in Theorem 3.5 should exist in reality even when some assumptions are not strictly satisfied.
>
> > Cumulative density function
>
> $F_X$ can have a large domain such as $\mathbb{R}$, but Theorem 3.5 only needs it to be convex in a **subset** $J$ of its domain. Formally, $\forall x_1, x_2 \in J , \lambda \in (0,1), f(\lambda x_1 + (1 - \lambda) x_2) \leq \lambda f(x_1) + (1 - \lambda) f(x_2)$. As an example, consider a 1-d Gaussian distribution $X \sim N(\mu, \sigma^2)$, we know it is convex if $J = (-\infty, \mu)$.
>
>
> > Continuity of $P(Y|X)$
>
> $P_{Y|X}(y|x)$ (where $y$ can be $0,1$) is the conditional probability for an agent with feature $x$ to have label $y$. This is a function of $x$ and we assume it is continuous in $x$. For example, the logistic function $P_{Y|X}(1|x) = \frac{1}{1+exp(-\beta^Tx)}$ is continuous when $x \in \mathbb{R}$.
>
> > $S_{m, t-1}$ in lines 187-188
>
> Yes, your understanding is correct. Because $S_{m, t-1}$ includes the model-annotated samples at $t-1$, the actual qualification of each sample is unobserved and is annotated using the pseudo-label generated by the model. In this sense, the "qualification rate" of $S_{m, t-1}$ is in fact the "acceptance rate." The reason we used "qualification rate" is because from the decision-maker's perspective, when updating its model using empirical risk minimization, it doesn't distinguish model-annotated samples from human-annotated ones and would regard pseudo-label as the actual qualification. In other words, the dataset the decision-maker uses to update the model has more and more fractions of people with positive labels (regardless of whether labels are acquired from humans or model), i.e., qualification rate of $S_{t}$ (augmented by $S_{m, t-1}$) is higher than that of $S_{t-1}$. We will add more clarification to the paper.
>
> ### References
>
> Levanon, S., & Rosenfeld, N. (2021, July). Strategic classification made practical. In International Conference on Machine Learning (pp. 6243-6253). PMLR.
>
> Ding, F., Hardt, M., Miller, J., & Schmidt, L. (2021). Retiring adult: New datasets for fair machine learning. Advances in neural information processing systems, 34, 6478-6490.

---

> > ### Comment · Reviewer_ovmu · 2024-08-09
> >
> > Dear Authors,
> >
> > Thank you for your response. I am currently reviewing your rebuttal and will have another reply to it. In the meantime, I am still unclear about the statement and conditions of Theorem 3.5. Could you please clarify the definition of "cumulative density function" as used in Theorem 3.5? It appears that the domain of this function is $\mathbb{R}^{d}$, given that it is assumed its restriction to a half-space in $\mathbb{R}^{d}$ is convex. If this is the case, could you also clarify what you mean by a function in $\mathbb{R}^{d}$ being decreasing, as mentioned in the proof of Theorem 3.5?
> >
> > Thank you,
> >
> > Reviewer

---

> > > ### Author Response · Authors · 2024-08-09
> > >
> > > Thanks for your timely reply!
> > >
> > > You are correct that "non-decreasing" in lines 957-958 means for each dimension $x_i$ in a $d$-dimensional feature $x \in J$, $F_X, P_X$ are non-decreasing. We will modify this sentence in the proof to make it clear. $F_X$ itself is non-decreasing in each dimension because it is CDF. Note that $P_X$ is the derivative of $F_X$, so the convexity itself ensures $P_X$ is non-decreasing in each dimension when $x \in J$.
> > >
> > > Thanks again for your careful review and suggestions.

---

> > ### Comment · Reviewer_ovmu · 2024-08-11
> >
> > Dear Authors,
> >
> > Thank you for your response. Based on your answer, I assume that when you work with the cumulative distribution function (not "cumulative density function" as mentioned in line 228 of your draft), you compute the CDF coordinate-wise. Also, when you say a function on $\mathbb{R}^{d}$ is non-decreasing, you mean it is non-decreasing in each coordinate. Please clearly state these conventions in your work, as they are not unique or standard assumptions about functions or distributions on $\mathbb{R}^{d}$. On the same line, regarding the assumption that $P_{Y|X}$ is continuous (mentioned in line 99), please specify that you mean $P_{Y|X}(1 \text{ or } 0 \mid x)$ is continuous over $x$. Otherwise, the way you have it there may confuse the readers. Additionally, in line 140, the domain of $h_{t}$ should be $\mathbb{R}^{d}$ (not $\mathbb{R}$ as you have it)? Please clarify.
> >
> > Assuming the above, I still have some clarifying questions regarding the conditions in Theorem 3.5. As I understand it, you are working with linear models, so $\mathcal{J}$ would be a subspace of $\mathbb{R}^{d}$ (as it is the decision boundary of a linear model). In your rebuttal above, you mentioned that $\mathcal{J}$ could be $(-\infty, \mu)$. However, it seems to me that $(-\infty, \mu)$ cannot arise in the context of Theorem 3.5, as $(-\infty, \mu)$ is not a subspace of $\mathbb{R}$. Am I missing something here?
> >
> > With that in mind, could you please elaborate on why Gaussian distributions or Beta distributions satisfy the convexity conditions in Theorem 3.5 (as $\mathcal{J}$ could be any subspace of $\mathbb{R}^{d}$)?
> >
> > Thanks,
> >
> > Reviewer

---

> > > ### Author Response · Authors · 2024-08-11
> > >
> > > Thanks for your follow-up. We address your concerns as follows.
> > >
> > > - We agree with your points in paragraph 1 and will be happy to modify our draft as you suggested. Also, you are correct on the definition of CDF and the domain of $h_t$, and we will correct the typo.
> > >
> > > - We kindly clarify that $J$ is not the **decision boundary** itself. In line 101, we define $f_t$ as the classifier from $\mathbb{R}^d \rightarrow$ {0,1} (not $[0,1]$) which directly maps the feature to the label. Therefore, $J =$ { $x|f_0(x) = 0$ } is actually the **half-space** (not a subspace) in $\mathbb{R}^d$ separated by the decision boundary as we write in Theorem 3.5.
> > >
> > > - With these clarifications, Gaussian and Beta distributions possibly satisfy Theorem 3.5 and we were using $1$-d cases as examples: (i) Gaussian example says when $f_0$ sets an admission threshold smaller than or equal to $\mu$, then the convexity assumption is satisfied because the CDF $\frac{1}{\sqrt{2\pi}}exp \frac{(x-\mu)^2}{2\sigma^2}$  is convex in $(-\infty, \mu)$ and $J \in (-\infty, \mu)$; (ii) Similarly, consider the CDF of the Beta distribution parameterized by $\alpha, \beta$, we can derive the sign of its second order derivative is the same as $(\alpha + \beta - 2) \cdot (\frac{\alpha-1}{\alpha+\beta-2} - 1)$. Then if $\beta = 1$ and $\alpha \ge \beta$, the second-order derivative will always be positive to ensure the convexity in its domain. Note that we do not say any Gaussian/Beta CDF will satisfy Theorem 3.5. Instead, these are examples that these distributions may satisfy Theorem 3.5.
> > >
> > > Based on our discussion, we are happy to emphasize that $f$ is the classifier again in Theorem 3.5 to avoid confusion and elaborate more on the examples (e.g., discuss the convexity of Gaussian/Beta CDF in more detail) in the Appendix.

---

> > > > ### Comment · Reviewer_ovmu · 2024-08-11
> > > >
> > > > Dear Authors,
> > > >
> > > > Thank you for the clarification. Let’s assume the half-space case as you mentioned. My other question still stands. I am particularly interested in higher-dimensional cases, as they are more relevant to real-world scenarios. For the sake of our discussion, we can restrict to $d=2$ or higher instead of $d=1$. Could you please comment on the conditions under which the convexity assumption of Theorem 3.5 holds for multivariate Gaussian or multivariate Beta distributions (as $\mathcal{J}$ could potentially be any half-space in $\mathbb{R}^{d}$)?
> > > >
> > > > Thanks,
> > > > Reviewer

---

> > > > > ### Author Response · Authors · 2024-08-12
> > > > >
> > > > > Thanks for your follow-up. We answer your question as follows.
> > > > >
> > > > > - For distributions with **convex** Cumulative Distribution Function in the entire 1-d domain (e.g., beta distribution with parameters $\beta = 1$ and $\alpha \ge \beta$, uniform distribution, exponential distribution), conditions for Theorem 3.5 always hold in multi-dimensional cases. For other distributions such as Gaussian, we admit that satisfying the conditions in Theorem 3.5 rigorously might be challenging in multi-dimensional cases. Consider a simple case of 2-d Gaussian $(x_1,x_2)$ with mean $(\mu_1, \mu_2)$ and $x_1$ and $x_2$ two independent Gaussian variables, the convexity holds when $x_1 < \mu_1$ and $x_2 < \mu_2$, i.e., convexity only holds on “half” of the half-space $J$, the condition in Theorem 3.5 does not hold.
> > > > >
> > > > > - Despite the difficulty of satisfying the conditions, we want to emphasize that these conditions are only sufficient but not necessary, which means the phenomenon shown in Theorem 3.5 might hold more easily in practice. Indeed, we provided sufficient empirical analysis to complement the theorem (as highlighted in line 239) which includes the situations you mentioned. Specifically, the Gaussian dataset used in experiments is high-dimensional, and the blue lines in Fig 5-6 show that the results in Theorem 3.5 hold well empirically; The features in the credit approval dataset conform Beta distributions which violate the convexity assumption, but Figure 11 in page 19 still shows a similar trend as theorems
> > > > >
> > > > > - In many practical scenarios of Strategic Classification, the decision-maker often aggregates agents’ features and makes decisions based on one-dimensional representation, and the agents directly change their features in that 1-d representation space. Typical examples include: (i) Large credit agencies such as FICO and Experian aggregate consumers’ features and then produce a single credit score, then consumers strive to increase the score; (ii) The standardized tests aggregate students’ abilities in different subjects and give a single score. Such scenarios have been widely used in previous literature (e.g., [14] and [61]). Under these settings, Theorem 3.5 can be regarded as applying to a 1-d situation and the convexity conditions are much easier to be satisfied rigorously.
> > > > >
> > > > > Finally, we hope to restate that our paper is the first to model the long-term impacts of model retraining with agent strategic feedback considering more practical scenarios with *model-annotated* samples. With a comprehensive analysis of welfare and fairness, we believe our findings can promote trustworthy machine learning and shed light on future work.
> > > > >
> > > > > Thanks again for your prompt reply. Please let us know if you have any other questions.
> > > > >
> > > > > Authors

---

> > > > > > ### Comment · Reviewer_ovmu · 2024-08-12
> > > > > >
> > > > > > Thank you for your response. I still have two questions related to your previous answer to better understand the limitations and applicability of your work.
> > > > > >
> > > > > > 1. Consider a distribution with a convex cumulative distribution function (CDF), and let $X$ and $Y$ be random variables with such a distribution. Could you please explain why the joint distribution of $(X,Y)$ satisfies the convexity condition of Theorem 3.5 in your work? I am particularly interested in the case where $X$ and $Y$ could be potentially correlated and possibly not identically distributed.
> > > > > >
> > > > > > 2. My initial impression was that the $d=1$ case might be less applicable to real-world scenarios. However, based on your above response, I have a follow-up question regarding this case. For example, consider a situation where FICO scores are used to decide loan approvals. Assuming each applicant has a single feature (their FICO score) and the classifier is a threshold on this score (as you work with linear classifiers), and all technical assumptions (e.g., convexity) are satisfied, could you please elaborate on what Theorems 3.5 and 3.3 imply from a strategic classification perspective? For instance, given that there is only one feature, how could it happen that an applicant's score is above the threshold but they are not qualified for the loan?
> > > > > >
> > > > > > Thanks,
> > > > > >
> > > > > > Reviewer

---

> ### Author Response · Authors · 2024-08-12
>
> Thanks for your reply. We apologize that our previous response regarding the high-dimensional case has some inaccurate statements. We present the precise analysis regarding all situations from 1-dimensional to high-dimensional ones and answer your questions.
>
> > Theorem 3.5
>
> 1. All our previous statements are precise for the **1-dimensional** setting where Gaussian, Beta, and Uniform distributions satisfy the conditions in Theorem 3.5;
>
> 2. Theorem 3.5 itself is precise for the high-dimensional settings. However, you are correct that the convexity of CDF for $X$ and $Y$ does not ensure the convexity of the joint CDF $(X, Y)$. Consider the simplest independent 2-dimensional distribution setting where $F_X, F_Y$ are convex in 1-d space, we can derive the Hessian of the joint CDF as:
>
> $\begin{bmatrix} F’’_X \cdot F_Y & F’_X \cdot F’_Y  \\\
> F’_X \cdot F’_Y & F_X \cdot F''_Y \end{bmatrix}$
>
> The Hessian is PSD if $ F_X''(x) \cdot F_X(x) \ge F_X’(x)^2$ and $ F_Y''(y) \cdot F_Y(y) \ge F_Y’(y)^2$ hold for any $(x,y) \in J$, which is equivalent to say $log(F_X)$ and $log(F_Y)$ are convex functions. Thus, features with log-convex CDFs will satisfy the conditions in Theorem 3.5 such as the Uniform CDF and any CDF in the form of $F(x) = \lambda e^{\lambda x}$ if $\lambda \ge 1$.
>
> 3. However, as we stated, Theorem 3.5 is a sufficient condition. In the proof of Theorem 3.5 under high-dimensional distribution settings with feature $(X_1, ..., X_d)$, we can derive the same results if the joint CDF $F$ is convex w.r.t each coordinate $X_i$ in the full domain of $X_i$. This means a multivariate feature distribution will result in a decreasing qualification rate if each feature has convex CDF and they are independent. This is to say, If each variable satisfies Beta distributions with $\alpha \ge \beta, \beta = 1$, the results in Theorem 3.5 still hold.
>
> 4. Regarding the situation where multiple features are dependent, the analysis can be increasingly challenging. However, in practice, the decision-maker often uses domain expertise and dimension-reduction methods (e.g., PCA) to do feature selection first, preventing severe violations of independence.
>
> > The FICO score example
>
> Theorem 3.3 and 3.5 in the FICO score example can be interpreted as follows: the decision-maker sets the score threshold to give loans, but the qualification means whether they can repay the loans and this is hidden. Theorem 3.3 illustrates that the decision-maker will set a lower score threshold when retraining goes on (e.g., from 700 to 600), but the qualification rate (i.e., the proportion of agents who will repay the loans) decreases. “An applicant’s score is above the threshold but they are not qualified for the loan” happens since the decision-maker lowers the acceptance threshold due to the misleading information it gets during the retraining. This phenomenon is one of our main discoveries that has not been studied in previous literature.
>
> > Proposed modifications of the manuscript
>
> Taking all the discussions together, we are happy to make the following modifications to the content related to Theorem 3.5: (i) add the convexity condition w.r.t each coordinate $X_i$ as mentioned above; (ii) add the FICO example in the appendix to illustrate the use cases in practice; (iii) Add more detailed discussions on whether different distributions in high-dimensional space satisfy Theorem 3.5.

---

> > ### Comment · Reviewer_ovmu · 2024-08-12
> >
> > Dear Authors,
> >
> > Thank you for your response. Please address the points mentioned earlier in our discussion in the revised draft to clarify the scope and applicability of the model for the readers.
> >
> > Despite some limitations in the theoretical results that are important to note (as previously discussed), I find the model to have some interesting features, particularly its dynamic approach to strategic classification and the incorporation of model-annotated samples. The topic of strategic classification and the deployment of ML models in social domains is indeed important, and I believe the work has the potential to benefit the community.
> >
> > Please clearly discuss the limitations and applicability of the model, so that these can be addressed/improved in future research. Consequently, I am raising my score from 4 to 5 and also increasing my confidence score.
> >
> > Thank you for your detailed and thorough responses during the discussion period, which helped me to better understand your work.
> >
> > Sincerely,
> > Reviewer

---

> > > ### Author Response · Authors · 2024-08-12
> > >
> > > Thanks for your endorsement and for increasing the score. We appreciate your suggestions and will revise the draft as highlighted in our discussion.

---

### Official Review · Reviewer_WrhT · 2024-07-13

**Soundness:** 3
**Presentation:** 3
**Contribution:** 3
**Rating:** 6
**Confidence:** 3

**Summary:**

This paper studies the effects of retraining machine learning (ML) models with data annotated by both humans and the models themselves, especially in social domains where human behavior is influenced by the ML systems. The authors explore how strategic human agents, who adapt their behaviors to receive favorable outcomes from ML models, can create feedback loops that affect the model's performance and fairness over time.

The paper begins by formalizing the interactions between strategic agents and ML models. It shows that as agents adapt to the decision policies of the model, they increasingly receive positive outcomes. However, this becomes problematic where the proportion of agents with truly positive labels may decrease over time. To address this issue, the authors propose a refined retraining process designed to stabilize these dynamics.

The authors analyze how fairness constraints imposed during each round of model retraining might impact disadvantaged groups in the long run. They find that enforcing common fairness constraints can sometimes fail to benefit these groups, highlighting the complexity of maintaining fairness in dynamic environments. The empirical section of the paper includes experiments on both semi-synthetic and real datasets. These experiments validate the theoretical findings, demonstrating that the proposed retraining process can mitigate some of the adverse effects of strategic behavior. The results show that while acceptance rates of agents tend to increase, the actual qualification rates may decrease under certain conditions, leading to a growing discrepancy between perceived and actual qualifications.

**Strengths:**

- The paper addresses a critical and relatively unexplored issue in ML – the long-term impacts of retraining models in the presence of strategic human behavior.

- The formalization of interactions between strategic agents and ML models provides a robust foundation for analyzing these dynamics.

- The investigation into how retraining processes affect algorithmic fairness is an important contribution of this work

- The use of semi-synthetic and real data to validate the theoretical findings strengthens claims of the study

**Weaknesses:**

- The theoretical analysis depends on several assumptions, such as the monotone likelihood ratio property and the availability of perfect information about agent distributions, which may not hold true in all real-world scenarios.

- The mathematical models and proofs are complex, potentially making it challenging for practitioners to implement the findings directly.

- While the experiments support the theoretical findings, the datasets used might not fully represent the diversity and complexity found in real-world applications.

**Questions:**

- Can you provide more empirical evidence or case studies to support the validity of the assumptions made in your theoretical analysis?

- Have you considered using more diverse and larger datasets to validate your findings? How might the results change with different types of data?

- How do you envision your findings being applied to other domains beyond college admissions and hiring? Are there specific adjustments needed for other applications?

**Limitations:**

- The study mainly focuses on specific social domains like college admissions and hiring, and its applicability to other fields remains uncertain.

- The proposed solutions, including the refined retraining process and fairness interventions, may not always be effective in practice. Real-world conditions can vary significantly from controlled experimental settings, which might affect the outcomes.

---

> ### Author Rebuttal · Authors · 2024-08-05
>
> Thanks for the comments. We address your questions point by point as follows.
>
> > Empirical evidence or case studies to support the validity of the assumptions
>
> - The monotonic likelihood ratio property (MLRP):
>
> This property is rather standard and has been widely used in literature (e.g., [11,12] in the main paper; Jung et al., 2020; Khalili et al., 2021; Barman et al., 2020); it means that each feature dimension is a good indicator of the target variable and an individual is more likely to be qualified when his/her feature value increases.
>
> Regarding the empirical evidence, we demonstrate the distribution of each of the two features in the Credit Approval dataset in Figure 10, where the fitted beta distribution satisfies the monotonic likelihood distribution. Moreover, [11] referred to in the main paper used FICO credit data and verified the monotonic likelihood assumption holds for people's FICO scores in Figure 15 of [11]. FICO has been widely used in the United States to assess people's creditworthiness and is a good practical example. In addition to the lending domain, MLRP has been studied in many other contexts, e.g., research has shown that higher wealth levels are often associated with higher likelihoods of certain favorable economic behaviors (like investment and consumption patterns), empirical data on insurance claims shows that higher levels of risk factors (like age, smoking status) are monotonically associated with higher likelihoods of claims.
>
> - The perfect information about agent distributions
>
> We want to clarify that the knowledge of agent distributions is only used when deriving theorems. In all the experiments, the decision-maker updates models by running empirical risk minimization on training data without knowing agent distribution $P_X$. To verify whether the conditions/assumptions in theorems hold, the decision-maker in practice may fit $P_X$ using the initial training set (e.g., Figure 10 for credit approval data).
>
> Last, we emphasize that the assumptions made in the theoretical analysis are sufficient but not necessary, i.e., the results may still hold even when assumptions are violated. This is validated in our experiments, where only the "uniform-linear" data satisfies all assumptions in the paper but all results are consistent with theorems.
>
> > Diverse/larger datasets and other domains
>
> - Our paper focuses on strategic classification settings [1,6,7,11,12,17,25,31,35]. According to the previous literature, human strategic behaviors are prevalent in high-stakes domains such as hiring, lending, and college admission, where humans have more incentives to improve their features for favorable outcomes. We believe the datasets adopted in our experiments are sufficiently representative and cover well-known data widely used in strategic classification literature.
>
> - We want to emphasize that our theoretical findings do not have any restriction on the dataset and are broadly applicable to other domains such as recommendation systems, fraud detection, etc., as long as they involve human strategic behavior and the decision-maker uses a linear model to make decisions. Note that the linearity of the decision model is a standard setting in strategic classification. In practice, one can first use a non-linear feature extractor to learn the embeddings of agents' preliminary features and then apply the linear model to the newly generated embedded features. Given the knowledge of both feature extractor and model, agents will best respond with respect to new features, and all our results still hold. The generalization to non-linear settings is also discussed in [13,28] in the main paper and [Levanon and Rosenfeld, 2021].
>
> - To further validate the above arguments, we provide an additional case study using ACSIncome data [Ding et al., 2022], which contains information on more than $150K$ agents. In this study, the goal is to predict whether a person has an annual income $> 50000$ based on $53$ features such as education level and working hours per week. Specifically, we consider a decision-maker who first learns 2-D embeddings from $53$ original features using a neural network and then regards the embedding as the new feature. A linear decision model is trained and used on these new features to make predictions. We divide the agents into 2 groups based on their ages. Similar to the credit approval data, we then fit Beta distributions on the 2 groups and then verify the monotonic likelihood assumption (Figure 1,2 in the attached pdf). We then plot the dynamics of $a_t, q_t, \Delta_t$ for both groups when the systematic bias is either positive or negative. The results show that similar trends still hold for this large dataset (Figure 3 in the attached pdf).
>
> ### Conclusion
> While our theoretical analysis relies on some assumptions and the results are mainly evaluated in domains of lending and admission, the assumptions and datasets are mostly standard and commonly used in strategic classification literature. Nonetheless, we want to emphasize that our results are broadly applicable to other datasets and applications that involve human strategic behavior.
>
> ### References
> Christopher Jung, Sampath Kannan, Changhwa Lee, Mallesh Pai, Aaron Roth, and Rakesh Vohra. Fair prediction with endogenous behavior. In Proceedings of the 21st ACM Conference on Economics and Computation, pages 677–678, 2020.
>
> Mohammad Mahdi Khalili, Xueru Zhang, Mahed Abroshan, and Somayeh Sojoudi. Improving fairness and privacy in selection problems. In Proceedings of the AAAI Conference on Artificial Intelligence, volume 35, pages 8092–8100, 2021.
>
> Barman, S. and Rathi, N. Fair cake division under monotone likelihood ratios. In Proceedings of the 21st ACM Conference on Economics and Computation, pp. 401–437, 2020.
>
> Ding, F., Hardt, M., Miller, J., & Schmidt, L. (2021). Retiring adult: New datasets for fair machine learning. Advances in neural information processing systems, 34, 6478-6490.

---

### Author Rebuttal · Authors · 2024-08-05

# Global Rebuttal

We thank the reviewers and AC for reviewing our paper. Here we present a global response to the questions shared by multiple reviewers.

## Assumptions and Applicability of Our Model

- The monotonic likelihood ratio property (MLRP)

This property is rather standard and has been widely used in literature (e.g., [11,12] in the main paper; Jung et al., 2020; Khalili et al., 2021; Barman et al., 2020); it means that each feature dimension is a good indicator of the target variable and an individual is more likely to be qualified when his/her feature value increases.

Regarding the empirical evidence, we demonstrate the distribution of each of the two features in the Credit Approval dataset in Figure 10, where the fitted beta distribution satisfies the monotonic likelihood distribution. Moreover, [11] referred to in the main paper used FICO credit data and verified the monotonic likelihood assumption holds for people's FICO scores in Figure 15 of [11]. FICO has been widely used in the United States to assess people's creditworthiness and is a good practical example. In addition to the lending domain, MLRP has been studied in many other contexts, e.g., research has shown that higher wealth levels are often associated with higher likelihoods of certain favorable economic behaviors (like investment and consumption patterns), empirical data on insurance claims shows that higher levels of risk factors (like age, smoking status) are monotonically associated with higher likelihoods of claims.

- The linearity assumption

The linearity of the decision model is a standard setting in strategic classification (e.g., [1,6,7,11,12,17,25,31,35]). In practice, one can first use a non-linear feature extractor to learn the embeddings of agents' preliminary features and then apply a linear model to the newly generated embedded features. Given the knowledge of both feature extractor and model, agents will best respond with respect to new features, and all our results still hold. The generalization to non-linear settings is also discussed in [13,28] in the main paper and [Levanon and Rosenfeld, 2021].

Finally, all assumptions made in our theorems are sufficient but not necessary, i.e., Theorem 3.5 may still hold even when assumptions are violated. This is mentioned in lines 238-239 and validated in experiments. Among all datasets adopted in experiments, only the "uniform-linear" dataset satisfies all assumptions in Theorem 3.5, while the Gaussian dataset satisfies Theorem 3.5 only if the initial qualification rate of agents is larger than 0.5. The German Credit and Credit Approval dataset only satisfies the linear classifier assumption and the monotonic likelihood assumption. However, the empirical results on all datasets demonstrate the validity of Theorem 3.5. Thus, we believe the phenomenon in Theorem 3.5 should exist in reality even when some assumptions are not strictly satisfied.

## Empirical Evidence of Our Model

-  The datasets: our paper focuses on strategic classification settings [1,6,7,11,12,17,25,31,35]. According to the previous literature, human strategic behaviors are prevalent in high-stakes domains such as hiring, lending, and college admission, where humans have more incentives to improve their features for favorable outcomes. We believe the datasets adopted in our experiments are sufficiently representative and cover well-known data widely used in strategic classification literature.

- **The additional case study**: To further validate the above arguments, we provide an additional case study using ACSIncome data [Ding et al., 2022], which contains information on more than $150K$ agents. In this study, the goal is to predict whether a person has an annual income $> 50000$ based on $53$ features such as education level and working hours per week. Specifically, we consider a decision-maker who first learns 2-D embeddings from $53$ original features using a neural network and then regards the embedding as the new feature. A linear decision model is trained and used on these new features to make predictions. We divide the agents into 2 groups based on their ages. Similar to the credit approval data, we then fit Beta distributions on the 2 groups and then verify the monotonic likelihood assumption (Figure 1,2 in the attached pdf). We then plot the dynamics of $a_t, q_t, \Delta_t$ for both groups when the systematic bias is either positive or negative. The results show that similar trends still hold for this large dataset (Figure 3 in the attached pdf).

### References

Christopher Jung, Sampath Kannan, Changhwa Lee, Mallesh Pai, Aaron Roth, and Rakesh Vohra. Fair prediction with endogenous behavior. In Proceedings of the 21st ACM Conference on Economics and Computation, pages 677–678, 2020.

Mohammad Mahdi Khalili, Xueru Zhang, Mahed Abroshan, and Somayeh Sojoudi. Improving fairness and privacy in selection problems. In Proceedings of the AAAI Conference on Artificial Intelligence, volume 35, pages 8092–8100, 2021.

Barman, S. and Rathi, N. Fair cake division under monotone likelihood ratios. In Proceedings of the 21st ACM Conference on Economics and Computation, pp. 401–437, 2020.

Ding, F., Hardt, M., Miller, J., & Schmidt, L. (2021). Retiring adult: New datasets for fair machine learning. Advances in neural information processing systems, 34, 6478-6490.

---

### Decision · Program_Chairs · 2024-09-25

**Decision:**

Accept (poster)

**Comment:**

This submission explores novel aspects in the area of strategic classification, namely the dynamics that develop when a population shifts due to strategic responses to published classification rules while the rules keep being adapted over time. All reviewers have found that the questions explored in this submission are novel and relevant. Initial questions raised were clarified during the author response period.

While the model and results provided in this submission have been found to be somewhat simplistic, the novelty of the questions asked will likely result in follow up studies on the initial set of results presented here. As such, this submission will make an interesting contribution to the program at NeurIPS.